# An automated high-resolution in vivo screen in zebrafish to identify chemical regulators of myelination

Jason J Early[1,2†], Katy LH Marshall-Phelps[1†], Jill M Williamson[1], Matthew Swire[1,3], Hari Kamadurai[4], Marc Muskavitch[4], David A Lyons[1,2]*

[1]Centre for Discovery Brain Sciences, University of Edinburgh, Edinburgh, United Kingdom; [2]United Kingdom Zebrafish screening facility, University of Edinburgh, Edinburgh, United Kingdom; [3]MRC Centre for Regenerative Medicine, Edinburgh, United Kingdom; [4]Biogen, Cambridge, United States

**Abstract** Myelinating oligodendrocytes are essential for central nervous system (CNS) formation and function. Their disruption is implicated in numerous neurodevelopmental, neuropsychiatric and neurodegenerative disorders. However, recent studies have indicated that oligodendrocytes may be tractable for treatment of disease. In recent years, zebrafish have become well established for the study of myelinating oligodendrocyte biology and drug discovery in vivo. Here, by automating the delivery of zebrafish larvae to a spinning disk confocal microscope, we were able to automate high-resolution imaging of myelinating oligodendrocytes in vivo. From there, we developed an image analysis pipeline that facilitated a screen of compounds with epigenetic and post-translational targets for their effects on regulating myelinating oligodendrocyte number. This screen identified novel compounds that strongly promote myelinating oligodendrocyte formation in vivo. Our imaging platform and analysis pipeline is flexible and can be employed for high-resolution imaging-based screens of broad interest using zebrafish.

**\*For correspondence:**
david.lyons@ed.ac.uk

[†]These authors contributed equally to this work

## Introduction

Myelin is a lipid-rich structure made by specialised glial cells, oligodendrocytes in the central nervous system (CNS) and Schwann cells in the peripheral nervous system (PNS), that facilitates the rapid propagation of nerve impulses and provides trophic and metabolic support to axons (*Nave and Werner, 2014*). In mammals, myelination in the CNS starts around birth and continues well into adult life (*Yeung et al., 2014*). New myelin is thought to be generated by a combination of the differentiation of new oligodendrocytes (*Hill et al., 2018*; *Hughes et al., 2018*; *Young et al., 2013*) and remodelling of existing myelin (*Auer et al., 2018*; *Snaidero et al., 2014*). New oligodendrocytes can be produced throughout life by tissue-resident oligodendrocyte progenitor cells (OPCs), which represent about 5% of CNS cells life-long (*Dawson et al., 2003*), and to a lesser extent by neural stem cell-derived OPCs (*Samanta et al., 2015*). There is increasing evidence that myelination is controlled by neuronal activity, and that regulation of myelin is important for both nervous system health and functional plasticity (*Almeida and Lyons, 2017*). Indeed, loss of myelin in late adulthood correlates with age-related cognitive decline (*Deary et al., 2006*; *Lu et al., 2013*; *Penke et al., 2010*). Reflecting its fundamental importance, it is increasingly clear that disruption to myelin and myelinating glia contributes to numerous diseases of the nervous system, including neuropathies of the PNS and a variety of conditions of the CNS. These include: neonatal hypoxia (*Buser et al., 2012*); childhood leukodystrophies (*Boespflug-Tanguy et al., 2008*); neurodevelopmental disorders including autism (*Pacey et al., 2013*) and schizophrenia (*Hercher et al., 2014*); as well as neurodegenerative conditions including multiple sclerosis (MS) (*Compston and Coles, 2008*), motor neuron disease

(*Kang et al., 2013*), Huntington's disease (*Huang et al., 2015*) and Alzheimer's disease (*Bartzokis, 2011*). Furthermore, disruption to oligodendrocytes and myelin play important roles in the pathogenesis of glioma (*Venkatesh et al., 2015*) and white-matter stroke (*Matute et al., 2013*). However, it is now clear that OPCs and neural stem cells present in the CNS throughout life can generate new oligodendrocytes following damage to myelin, and that these cells can in turn regenerate myelin through the process of remyelination (*Franklin and ffrench-Constant, 2017*; *Prineas and Connell, 1979*; *Samanta et al., 2015*; *Zawadzka et al., 2010*). This capacity to generate myelin throughout life, including in response to damage, raises hope that the oligodendrocyte lineage may be tractable for the treatment of disease.

Depending on the specific disease, multiple bottlenecks can limit the normal formation or regeneration of myelinating oligodendrocytes, including disruption to OPC specification, proliferation, migration, oligodendrocyte differentiation or (re)myelination, reviewed by (*Cole et al., 2017*). For example, in MS, while certain areas with demyelinated axons completely lack OPCs, others have plentiful OPCs that do not differentiate into oligodendrocytes (*Boyd et al., 2013*; *Chang et al., 2000*; *Kuhlmann et al., 2008*; *Lucchinetti et al., 1999*). Even when oligodendrocytes differentiate and regenerate myelin, new myelin sheaths are characterised by being shorter and thinner than normal (*Franklin and ffrench-Constant, 2017*; *Prineas and Connell, 1979*). The fact that many demyelinated lesions contain OPCs that do not differentiate has prompted efforts to identify strategies to promote oligodendrocyte differentiation. Indeed, recent unbiased phenotypic screens have led to the identification of several hit compounds that promote oligodendrocyte differentiation in vitro (*Deshmukh et al., 2013*; *Joubert et al., 2010*; *Mei et al., 2014*; *Mei et al., 2016*; *Najm et al., 2015*; *Peppard et al., 2015*; *Porcu et al., 2015*). A number of these hits have been validated in vivo (*Cole et al., 2017*), and one, clemastine, has even shown efficacy in improving optic nerve function in patients with MS (*Green et al., 2017*). This remarkable progress highlights the power of discovery screens, and indicates that the oligodendrocyte lineage is therapeutically tractable for the treatment of disease. However, there remains a great need to identify safe, effective, and specific compounds capable of regulating distinct stages of the oligodendrocyte lineage, from specification through to (re)myelination.

In recent years zebrafish have become well established for the study of oligodendrocyte biology and myelination in vivo, reviewed in (*Czopka, 2016*). Phenotypic gene discovery screens have identified novel factors required for myelination (for example *Kearns et al., 2015*; *Lyons et al., 2009*), and live imaging based studies have provided insight into cellular mechanisms of oligodendrocyte development and myelination (for example *Almeida et al., 2011*; *Auer et al., 2018*; *Czopka et al., 2013*; *Kirby et al., 2006*; *Snaidero et al., 2014*). Zebrafish are now also widely used for chemical biology and drug discovery, due to their conservation of molecular and cellular function with mammals and natural properties that make them ideal for scalable and efficient compound-based in vivo screens. Zebrafish embryos can be generated in very large numbers and their externally fertilised eggs develop rapidly from a single cell to a freely swimming, optically transparent animal with extensive myelination in just a few days, meaning entire developmental events and biological processes can be observed in real time (for example *Keller et al., 2008*). Compounds can be taken up by the embryo following direct addition to the water, and phenotypic screens can be carried out either by direct assessment of organ formation, cardiovascular function, general development or behaviour, or using a variety of cell-type and tissue specific markers and reporters (for example *Milan et al., 2003*; *North et al., 2007*; *Peterson et al., 2000, 2004*; *Rihel et al., 2010*). Since the first ground-breaking phenotypic chemical screen using zebrafish in 2000 (*Peterson et al., 2000*), numerous others have followed (*Rennekamp and Peterson, 2015*), including assessment of oligodendrocyte lineage formation (*Buckley et al., 2010*) and a number that have led to new clinical trials (*MacRae and Peterson, 2015*; *North et al., 2007*; *Yu et al., 2008*). Using an in vivo system is particularly advantageous for screening complex biological processes, such as myelination, that are difficult to fully recapitulate in vitro (*Buckley et al., 2008*), and recent advances in automated embryo handling technologies have made scalable in vivo screens using zebrafish feasible (*Chang et al., 2012*; *Pardo-Martin et al., 2010*; *Wang et al., 2015*). Here we combine a commercially available platform that automates the delivery of zebrafish from multi-well plates to a microscope platform, with a customised spinning disk confocal microscope to automate high-resolution image acquisition. We develop an image handling and analysis pipeline that allowed us to execute an automated screen which identified several novel hit compounds that regulate myelinating oligodendrocyte

number. In addition, we expect our imaging and analysis capacities to be broadly useful to researchers interested in executing scalable, high-resolution in vivo analyses of diverse biological processes in zebrafish.

## Results

### Automated imaging of zebrafish larvae at high resolution

In order to establish an automated high-resolution in vivo screening platform for zebrafish, we decided to couple the Large Particle (LP) Sampler and the VAST BioImager (*Pardo-Martin et al., 2010*) (both Union Biometrica, Holliston, MA) with a customised spinning disk confocal microscope (Carl Zeiss Microscopy GmbH, Jena, Germany), which together we refer to as VAST-SDCM. The LP Sampler automates the delivery of zebrafish larvae from the wells of a multi-well plate to the VAST BioImager platform (*Chang et al., 2012*). Core to the VAST BioImager is a thin-walled glass capillary (10 μm thick glass) in which zebrafish can be imaged (*Figure 1A–C*). The capillary can be rotated fully (360 degrees) around the x-axis (*Figure 1C,D*; *Video 1*), such that animals can be automatically oriented according to user-defined templates (*Pardo-Martin et al., 2010*). We decided to use a spinning disk confocal system for image acquisition, because of the balance of high-speed and high-resolution imaging capacity with the flexibility to customise the configuration on a standard upright microscope (see Materials and methods). The VAST BioImager can accommodate glass capillaries of 600–750 μm internal diameter, which allows imaging of zebrafish embryos and young larvae (*Figure 1C,D*).

*Table 1* provides an overview of image acquisition parameters and performance outputs, including acquisition speeds, resolution, and general throughput of the VAST-SDCM. In initial tests using our transgenic reporter of myelinating oligodendrocytes, Tg(mbp:EGFP), we found that we could image all of the myelinating glial cells in a zebrafish larva at 4 days post fertilisation (dpf), with a success rate of 83%, in approximately 60 s, using a 10X objective (849 nm pixel size) (*Figure 1E,F* and *Video 2*). In addition, we imaged Tg(mbp:EGFP-CAAX) larvae, in which EGFP is membrane localised, and therefore enriched in myelin, using a 20X objective (141 nm pixel size), highlighting the potential of the system for higher resolution imaging (*Figure 1G*).

Given the performance obtained in our initial studies, we reasoned that the VAST-SDCM platform could provide the speed and resolution required for scalable imaging-based screens of myelinating oligodendrocyte development in vivo.

### Image handling: selection of regions of interest

We first wanted to identify a region of the CNS that might be suited to an automated imaging-based screen. Myelinating oligodendrocytes differentiate in the zebrafish spinal cord from around 2 dpf, first in the ventral area and soon thereafter in the dorsal area, with significant numbers present by 4 dpf (*Figure 1E,F*). Due to their sparse distribution, we found that myelinating oligodendrocyte number in the dorsal spinal cord could be reliably assessed by compressing 3D z-stacks into 2D maximum intensity projections (MIPs): manual counts of cells in 2D MIPs average 33.3 vs 3D stereological counts of 35.4 (Student's t-test, p=0.61, *Figure 2—figure supplement 1*). To develop an automated method to quantify myelinating oligodendrocyte number in this region, we first needed a method to automatically select the dorsal spinal cord as a specific region of interest (ROI) in 2D MIPs. We applied a landmark-based segmentation approach to achieve this. The early myelination of the ventral spinal cord is reflected by high-intensity fluorescent reporter expression in Tg(mbp:EGFP) animals. This allowed us to automatically plot points of peak fluorescence in the X-Y plane of tiled 2D MIPs and draw a line between points and thus delineate myelinating oligodendrocytes of the ventral spinal cord (*Figure 2A–E*, *Video 3*). From this defined line, we plotted the distance to the dorsal-most cells in the spinal cord, which allowed us to demarcate the dorsal spinal cord as a specific area of interest (*Figure 2F,G*). Our image analysis script also allows users to manually adjust any occasional errors that may occur in defining the line demarcating the ventral spinal cord (referred to as 'Refined ROIs'). However, following the fully automated ROI specification process in 196 animals, we counted on average only 1.2 cells per animal outwith the automatically assigned ROIs (*Figure 2—figure supplement 2*). Therefore, we reasoned that we could use our 2D landmark-based segmentation process to work towards a fully automated screen to identify compounds that regulate myelinating

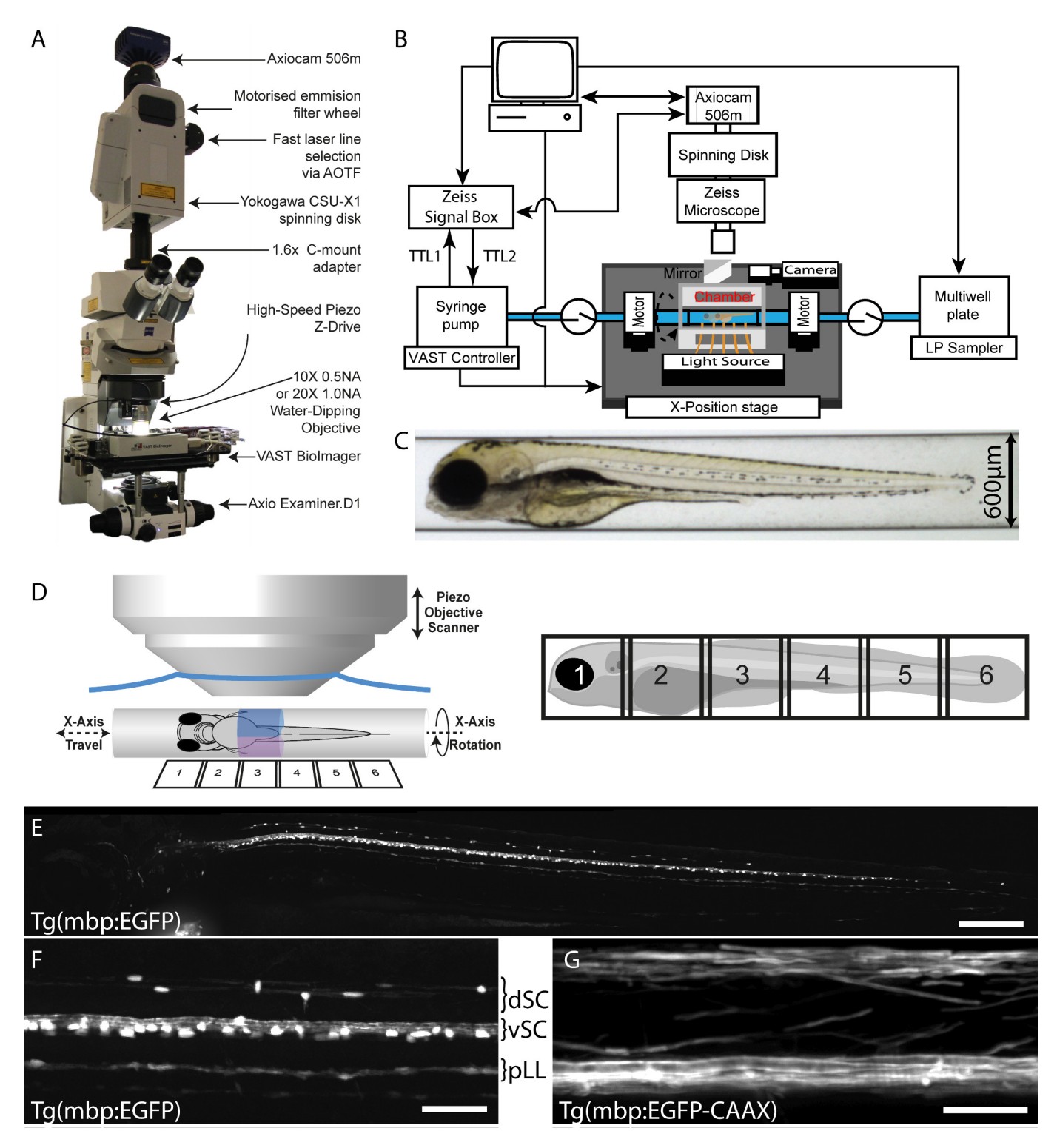

**Figure 1.** VAST-SDCM setup for rapid, automated image acquisition at subcellular resolution. (**A**) Overview of custom VAST BioImager and spinning-disk microscope setup, highlighting key components. (**B**) Schematic representation of hardware interaction between Zeiss SDCM, VAST BioImager and LP Sampler. (**C**) Representative brightfield image produced by VAST BioImager, showing a 4 dpf zebrafish embryo in a 600 µm glass capillary. (**D**) Illustration of relative setup of microscope objective and capillary. Note that the piezo range (400 µm) requires for two stacks (blue and magenta) to be acquired per tile of fish (6-tiles with 10% overlap required to image entire 4 dpf larva). (**E**) Maximum intensity projection of images of Tg(mbp:EGFP) zebrafish, following tiled merging of 6 z-stacks. Scale bar = 200 µm. (**F**) and (**G**) Example images of mid-trunk region of Tg(mbp:EGFP) and Tg(mbp:
*Figure 1 continued on next page*

*Figure 1 continued*

EGFP-CAAX) fish imaged with 10 × 0.5 NA (3 × 3 binning) and 20 × 1 NA (1 × 1 binning) objectives respectively. Scale bars = 50 and 25 μm respectively). dSC = dorsal spinal cord, vSC = ventral spinal cord and pLL = posterior lateral line.

oligodendrocyte number in the dorsal spinal cord. Whilst in this case we were able to successfully quantify our cells of interest in 2D MIPs, this is not always the case. As such, further to the above tools, we provide a method for selecting a 3D sub-region from an image dataset (3D_Crop_v1.1.ijm, see Materials and methods, *Figure 2—figure supplement 3A–C*, *Video 4*, and *Early, 2018*), or alternatively, a plugin for recording the depth of maximum intensity for a given xy location in a 3D dataset (D_Projector_v1.1.jar, see Materials and methods, *Figure 2—figure supplement 3D* and *Early, 2018*). These allow for either removal of unwanted/non-specific fluorescence, or recording of 3D spatial information, and could be combined with the scripts presented herein.

## Automated quantification of oligodendrocyte cell number

Having established a method to automatically demarcate the dorsal spinal cord as a specific ROI, we next tested the ability of an available 2D maxima identification tool (ImageJ's inbuilt findMaxima function) to identify cells in this region. To remove false positive maxima that do not resemble cells, we included an additional processing step called processCells, which can be used to assess cell size, shape and fluorophore expression (*Figure 3* and see Materials and methods). These functions were applied within the automatically identified dorsal ROI, and accurately quantified cell number compared to manual assessment (*Figure 3* and see Materials and methods). Our image handling script also saves the selections made by the findMaxima and processCells functions for posthoc validation and further analyses. For example, in circumstances where one may wish to manually confirm accuracy of cell counts, our script also allows for manual refinement of assignments (*Figure 3*). Manual refinement does come with a significant cost in investigator time. For example, the completely automated process can process an individual sample in under a second and the 288 animals in a 96 well plate in 3.5 min. In contrast manual refinement takes an average of 50 s per sample to process, and circa 4 hr per plate. Nonetheless this is still significantly less time than it takes for an expert to assess cell number in the dorsal spinal cord of 288 animals from 3D stacks (circa 34 hr). In summary, the combination of automated ROI specification and automated cell counting indicated that we could, in principle, execute a fully automated image acquisition and analysis screen for compounds that regulate oligodendrocyte development using the VAST-SDCM (*Video 5*).

## Validation of automated analysis to identify changes in myelinating oligodendrocyte number

Following a manual screen to identify compounds that regulated myelination, using our Tg(mbp: EGFP-CAAX) reporter, we identified the compound SKP2-C25 as promoting myelination. In this manual screen we treated embryos with compounds from 2 to 4 dpf and assessed myelination at 4 dpf. Follow-up analyses showed that SKP2-C25 increased myelination, as assessed by electron microscopy, and did so by increasing oligodendrocyte number (*Almeida et al., 2018*). SKP2-C25 is reported to be a selective inhibitor of the Skp2 SCF ubiquitin ligase complex that contributes to the ubiquitin-directed proteasomal degradation of proteins (*Chan et al., 2013*). To exemplify the power of zebrafish to define how compounds affect specific stages of oligodendrocyte lineage progression, we show here, using time-lapse microscopy that treatment of animals with SKP2-C25 increases the proliferation of OPCs between 2 and 3 dpf (*Figure 4—figure supplement 1* and *Video 6*). We have previously shown that oligodendrocytes differentiate by default in zebrafish to mediate the early myelination observed in the larval spinal cord (*Almeida et al., 2018*). Therefore, upon an increase of OPC proliferation following SKP2-C25 treatment, there is a larger pool of OPCs that differentiate by default, leading to the observed increase in myelinating oligodendrocytes.

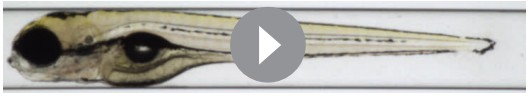

**Video 1.** Movie of 4 dpf zebrafish larva in the VAST BioImager capillary, showing the x-axis 360° rotational capacity. Refers to *Figure 1*.
https://elifesciences.org/articles/35136#video1

**Table 1.** Imaging and Screening Statistics.

Exemplar resolution and imaging times for given pixel binning levels, using a 10 × 0.5 NA objective and imaging our Tg(mbp:EGFP) reporter line. Fish loading times by LP Sampler to VAST BioImager based on three fish per well (experiment averages 52 s for first fish and 17 s for subsequent two per well). Combined loading and imaging rates per fish for different imaging parameters.

| Pixel binning | 1 × 1 | 2 × 2 | 3 × 3 |
|---|---|---|---|
| Pixel Size (nm) | 283 | 566 | 849 |
| Exposure (ms) | 180 | 45 | 20 |
| Approximate Imaging Time (s/fish) | 260 | 80 | 60 |
| Fish loading time (s/fish) | 30 | 30 | 30 |
| Load and Image Rate (fish/hour) 5 tiles | 12 | 33 | 40 |
| Load and Image Rate (fish/hour) 1 tiles | 43 | 78 | 85 |

Given that SKP2-C25 increased myelinating oligodendrocyte number, we decided to use this compound to assess the sensitivity of our automated imaging and analysis pipeline to detect relatively subtle changes in oligodendrocyte number. We imaged control animals and those treated with a range of concentrations of SKP2-C25 using the VAST-SDCM, and found that our imaging and analysis pipeline was indeed able to detect increases in myelinating oligodendrocyte number (*Figure 4*). Therefore, we reasoned that our imaging and analysis pipeline had the sensitivity to detect changes in myelinating oligodendrocyte cell number and thus could be employed to execute a fully automated screen.

## Chemical screen for compounds that regulate myelinating oligodendrocyte number

Having established a robust automated method for quantifying oligodendrocyte number in vivo, we next wanted to carry out a screen for regulators of myelinating oligodendrocyte formation. For this, we collated a small focused library of 175 compounds from three commercial sources (146 distinct molecular entities) that targeted enzymes implicated in epigenetic and post-translational modifications, given the known importance of epigenetic mechanisms during oligodendrocyte development (*Copray et al., 2009*; *Liu et al., 2016*). We reasoned that using a library with such diverse targets would allow us to identify compounds affecting distinct stages of oligodendrocyte lineage progression from OPC specification through to differentiation and myelination. To ensure that our screen would detect relatively subtle changes in oligodendrocyte number, we calculated that the sample size needed to detect the increase in cell number seen for SKP2-C25 with 95% power and p<0.05 significance values was 9 (see Materials and methods). To account for potential automation errors (our initial data indicated an 83% success rate for accurate loading, plus subsequent positioning and imaging), the number of animals plated for analysis was increased to 12. Prior to our imaging-based screen, we first carried out an assessment of compound toxicity by treating animals from 2 to 5 dpf with all compounds at 100 µM, 10 µM and 1 µM in 96 well plate format (see Materials and methods). We found that 59/175 compounds were lethal at 100 µM, 22/175 at 10 µM, and 0/175 at 1 µM. Given that SKP2-C25 was effective in increasing oligodendrocyte number at 2 µM, and the absence of any lethal compounds in our library at 1 µM, we decided to

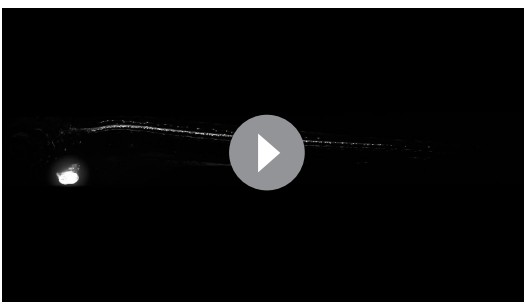

**Video 2.** Movie of maximum intensity projection of stitched confocal stacks showing a 4 dpf Tg(mbp:EGFP) animal demonstrating the ability of the VAST-SDCM system to image myelinating glia at subcellular resolution throughout the whole animal. Images acquired at 10X, using 3 × 3 binning, pixel size 849 nm. Refers to *Figure 1*.
https://elifesciences.org/articles/35136#video2

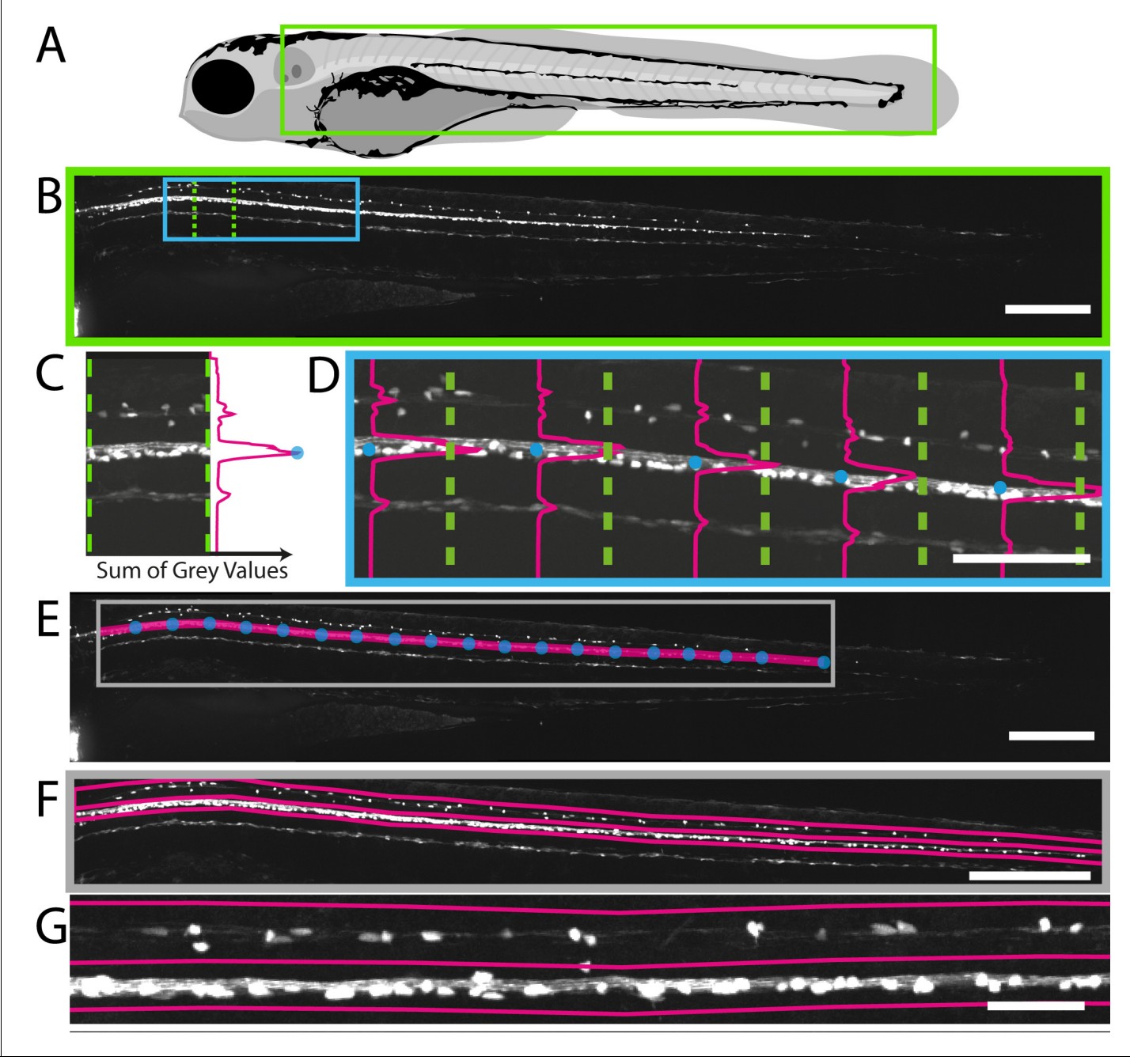

**Figure 2.** Fully automated ROI specification in Tg(mbp:EGFP) larval zebrafish. (A) Schematic overview of 4 dpf larvae, illustrating the region (green rectangle) of Tg(mbp:EGFP) animal imaged using VAST-SDCM (B). (C) Example fluorescence intensity profile from region demarcated by green dashed lines in B. Each point on the profile is a sum of the grey values in X for a given Y position between the green dashed lines. (D) Along the length of the spinal cord, each point of peak fluorescence is marked by a blue circle. The points identified in this way delineate the ventral spinal cord (E). This line was then used to set regions of interest for dorsal and ventral spinal cord (F) and higher magnification view in G). Scale bars B, E and F = 250 μm, D = 100 μm, and G = 50 μm.

The online version of this article includes the following figure supplement(s) for figure 2:

**Figure supplement 1.** Comparison of manual 3D vs manual 2D cell counting methods.
**Figure supplement 2.** Analysis of plotROI function.
**Figure supplement 3.** 3D-Cropping of images of complex transgenic animal.

carry out our screen of myelinating oligodendrocyte number at 2 µM.

For our screen, Tg(mbp:EGFP) embryos were dechorionated and arrayed, three per well, into 96-well plates at 1 dpf. Animals were treated with 2 µM compound at 2 dpf for 2 days (2–4 dpf), covering the developmental window in which OPCs are first specified, migrate and differentiate into oligodendrocytes that form myelin. At 4 dpf, larvae were imaged using the VAST-SDCM platform (see above and Materials and methods) and effects on myelinating oligodendrocyte number were quantified using our fully-automated processes (*Figure 5A* and Materials and methods). Effects of compound treatments on myelinating oligodendrocyte number were normalised against matched DMSO-treated controls for each screening session and ranked by increase in cell number (*Figure 5B* and *Supplementary file 1* and *2*). Of the 146 unique compounds tested, 14 were identified as having a significant effect on oligodendrocyte number. Statistical analysis of individual experiments (from single clutches) identified five compounds as significantly increasing cell number, including the positive control SKP2-C25, and nine as decreasing, all assessed by Ordinary one-way ANOVA with Dunnett's post hoc multiple comparisons test (see Materials and methods and *Table 2*). Our positive control compound was included in two independent screening runs, but only returned as a statistically significant hit in one of those runs. Interestingly, the screening run in which it was not identified as a hit took twice as long to complete (8 vs 4 hr), reflecting the fact that variability in cell numbers will increase with time, given ongoing development of live animals. Two compounds (ML-228 and Trichostatin A) resulted in developmental lethality at the screening concentration used (*Supplementary file 1* and *2*). To confirm that our fully automated image analysis was accurate and did not itself lead to the creation of false positives or negatives, we compared the results from a single screening session with manually-verified counts where the user could correct for any inaccurate spinal cord selection or oligodendrocyte quantification (*Figure 5C* and *Supplementary file 3*). This comparison demonstrated that the automated and manually adjusted analyses revealed essentially identical effect sizes and identified the same hit compounds (*Figure 5C* and *Supplementary file 3*).

In agreement with previous literature on the critical role of histone deacetylation for oligodendrocyte development (*Cunliffe and Casaccia-Bonnefil, 2006*; *Liu et al., 2016*), we found several HDAC (histone deacetylase) inhibitors in our set of 9 compounds that significantly reduced oligodendrocyte cell number (*Figure 5—figure supplement 1C and D*, *Table 2* and *Supplementary file 1* and *2*). The four novel compounds that significantly increased numbers of MBP-expressing oligodendrocytes in vivo (splitomicin, C646, NU9056, and GSK-J5), have activity on diverse apparent targets. Of these, splitomicin, a sirtuin histone deacetylase inhibitor (*Bedalov et al., 2001*), and C646, a p300/CBP histone acetyltransferase inhibitor (*Bowers et al., 2010*), were the most effective in enhancing oligodendrocyte number, both demonstrating a 73% increase compared to controls (DMSO 41.7 ± 12.5 vs 2 µM splitomicin 72.3 ± 21.2 MBP +ve oligodendrocytes and DMSO 34.3 ± 10.1 vs 2 µM C646 59.1 ± 14.3; both p=0.0001; *Supplementary file 1* and *2*). NU9056, a selective KAT5 (Tip60) histone acetyltransferase inhibitor (*Coffey et al., 2012*), increased oligodendrocyte number by 56% (DMSO 38 ± 18.9 vs 2 µM NU9056 59.2 ± 12.6, p=0.013; *Supplementary file 1* and *2*). GSK-J5, which shows weak inhibition for the jumonji family of histone demethylases, (*Kruidenier et al., 2012*) increased myelinating oligodendrocyte number by 41% (DMSO 38.4 ± 9.7 vs 2 µM GSK-J5 54.2 ± 9.1, p=0.003). As our image acquisition pipeline also acquires bright-field images of each larvae, one can immediately also assess whether compounds give rise to morphological phenotypes or developmental toxicity (see examples in *Figure 5—figure supplement 1*).

To assess the concentrations at which our four novel hit compounds functioned, and to test the possibility that we may have identified false positives, we carried out concentration series analyses of hits at 0.5, 1, 2, 4, 10 and 20 µM using the VAST-SDCM and refined ROI and refined counts analysis. All four compounds were lethal at 10

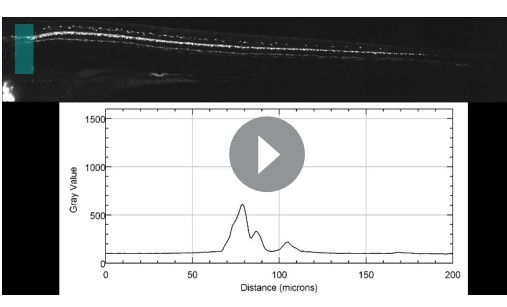

**Video 3.** Movie showing the automated process by which fluorescent intensity profiles are calculated at defined positions along the spinal cord of a 4 dpf Tg(mbp:EGFP) animal to plot the position of the ventral spinal cord. Refers to *Figure 2*.
https://elifesciences.org/articles/35136#video3

**Video 4.** Movie showing the principle of the 3D cropping macro, starting with travelling through the z planes of a 3D dataset acquired on the VAST-SDCM of a 4dpf Tg(mbp:nls-EGFP) animal. This is followed by a maximum intensity projection through the Z-axis, with the region to be kept highlighted in cyan. The projection is then rotated through 90 degrees, to give the Y-axis maximum intensity projection, which then has the region to be retained highlighted in cyan. The macro then removes any pixels not contained within both regions to give the final 3D dataset, which is then rotated back to the original orientation. Refers to *Figure 2—figure supplement 3*.
https://elifesciences.org/articles/35136#video4

and 20 µM. NU9056 re-treatment did not demonstrate an increase in cell number at any concentration (*Figure 6*), suggesting the possibility that its identification in our screen represented a false positive. However, splitomicin, C646 and GSK-J5 recapitulated the observed increases in oligodendrocyte number identified in the screen (*Figure 6*).

In summary, our automated screening pipeline represents an efficient, sensitive and unbiased approach for the identification of novel compounds that can promote myelinating oligodendrocyte formation.

## Automated region-specific analyses to improve screening throughput

To extend the functionality of our image analysis pipeline and to further characterise the effects of our hits, we wanted to assess whether the compounds identified in our screen elicited any region-specific effects on myelinating oligodendrocyte formation. Therefore, we adapted our automated image analysis script to record the x, y coordinates of each identified MBP +ve cell within the dorsal spinal cord ROI and to plot the distribution of oligodendrocytes along the anterior to posterior (A-P) axis in 100 µm bins (see Materials and methods and *Figure 7B and D*). This method identified a greater proportion of mbp:EGFP expressing oligodendrocytes in the anterior region of the control spinal cord, reflecting the known A-P gradient of oligodendrocyte development. As per our concentration series, we found that retreatment of animals with NU9056 at its highest tolerated dose (4 µM) failed to show a significant increase in myelinating oligodendrocyte number, when all cells in the dorsal spinal cord were assessed (*Figure 7C*). However, we found that NU9056 treatment caused a clear increase in oligodendrocyte number in the anterior spinal cord, but no discernible effect in the posterior spinal cord, which was also true of GSK-J5 (*Figure 7B*). In contrast, splitomicin, C646 and SKP2-C25 exhibited a consistent increase in cell number along the A-P axis (*Figure 7D*). Our region-specific assessment of hit compounds indicated that the region representing the second of five tiles along the length of the spinal cord was the most sensitive to changes in myelinating oligodendrocyte number, wherein all five compounds had significant increases in myelinating oligodendrocyte number (*Figure 7E* and *Supplementary file 3*). These data suggest that analyses in single tile mode would be sufficient to detect hit compounds regulating cell number for future screens. The region-specific effects may reflect timing of the A-P gradient of oligodendrocyte lineage specification through to myelination with respect to the timing of our treatment (2 to 4 dpf), rather than a region-specific effect per se. Imaging one instead of 5-tiled z-stacks per animal would more than double the throughput of future screens, and allow us to assess myelinating oligodendrocyte number in 85 animals per hour, instead of the initial screen throughput of 40 per hour (*Table 1*).

## Automated analysis of myelination

The myelin sheaths that are formed by oligodendrocytes during remyelination are abnormally short and thin (*Franklin and ffrench-Constant, 2017*; *Prineas and Connell, 1979*). Therefore, the identification of compounds that promote myelination per se may be important in restoring optimal neural circuit function via myelin regeneration (*Cole et al., 2017*). To develop an assay to automatically assess extent of myelination, we made use of our transgenic reporter line Tg(mbp:EGFP-CAAX), in which EGFP is targeted to the membrane of MBP-expressing oligodendrocytes (*Figure 8A*; [*Almeida et al., 2011*]). To quantify changes in myelination, we first applied our landmark-based segmentation approach to select the dorsal region of the spinal cord, as above for Tg(mbp:EGFP). After ROI selection, we used ImageJ's 'getRawStatistics' function to quantify the average fluorescence intensity throughout the segmented region, as a proxy for the extent of myelination (see Materials and methods). We first tested whether this assay could identify the known increase in

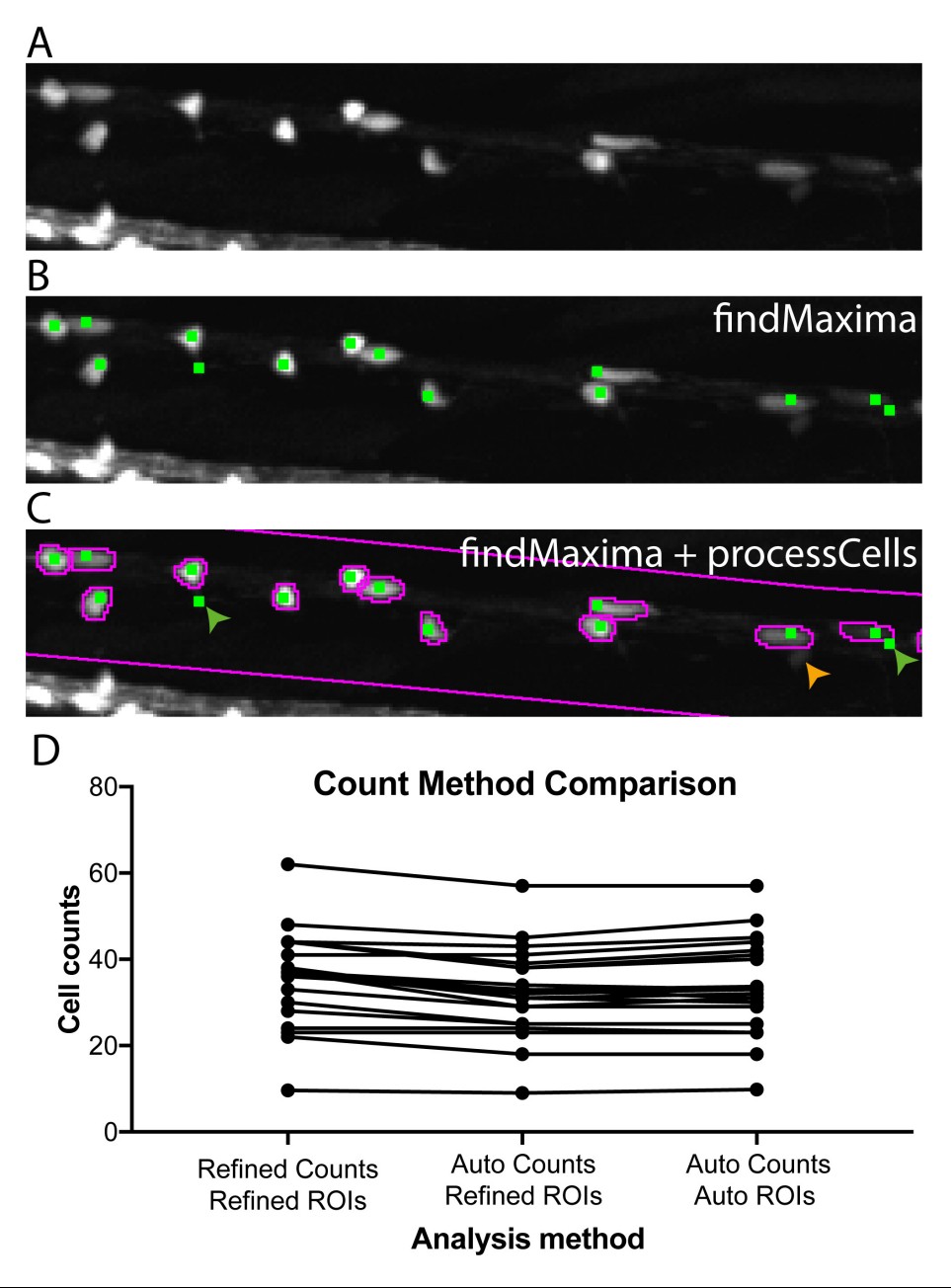

**Figure 3.** Automated cell counting allows for effective quantification of oligodendrocyte cell number. (**A**) Crop of dorsal spinal cord of 4 dpf Tg(mbp:EGFP) zebrafish larva acquired on VAST-SDCM. (**B**) Output of ImageJ's findMaxima function (green dots), note that whilst not every cell is identified (e.g. the small dim cell indicated by orange arrow in C), some cells are identified multiple times (green arrows in C). (**C**) Output of findMaxima plus processCells function allows assessment of cell likeness, and removal of false positives, for example green arrows. (**D**) Comparison of count methods for individual fish, with ROIs and cell counts verified and corrected by a human, ROIs corrected by a human with automatic cell counts, and fully automated ROIs and cell counts.

myelination over time and indeed found significant increases in mean fluorescence intensity in the dorsal spinal cord of our Tg(mbp:EGFP-CAAX) from 3-4 dpf and 4–5 dpf, respectively (average dorsal grey values – 1 × 1 binning, 80 ms exposure, 3 dpf: 4.91 ± 0.68; 4dpf: 19.1 ± 6.41, p<0.0001 (3 vs 4 dpf); 5 dpf: 33.28 ± 11.44, p<0.0001 (4 vs 5 dpf); *Figure 8—figure supplement 1*). We next wanted to investigate if our hit compound splitomicin exhibited an increase in myelination,

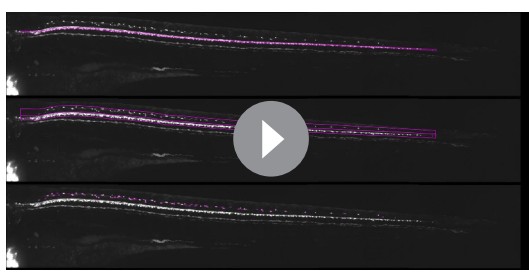

**Video 5.** Movie showing circa real-time speed of automated ventral spinal cord demarcation, dorsal spinal cord ROI selection and cell counting per animal. Refers to *Figures 1–3*.
https://elifesciences.org/articles/35136#video5

according to this assay. Automated quantification of splitomicin-treated larvae revealed a significant increase in mean dorsal fluorescence intensity over multiple concentrations compared with control-treated larvae, despite this being difficult to subjectively appreciate by eye (average dorsal grey values – 3 × 3 binning 60 ms exposure, DMSO 136.1 ± 35.9 vs 2 µM splitomicin 174.7 ± 34.3, p=0.0023; 3 µM splitomicin 186.4 ± 33.3, p=0.0001; 4 µM splitomicin 203.4 ± 38.3, p=0.0001 *Figure 8B*). To confirm that this increase in fluorescence did indeed reflect a change in bona fide myelination, we performed transmission electron microscopy on 4 dpf larvae and found a ~1.8 fold increase in myelinated axon number in splitomicin-treated animals (DMSO 9.8 ± 4.3 vs 2 µM splitomicin 18 ± 2.6, p=0.0065; *Figure 8C–E*). Therefore, our fluorescence intensity-based assay provides an unbiased, automated approach for the quantification of myelination in vivo, with higher sensitivity than subjective manual assessment. Future screens will be carried out to identify compounds that can promote myelination in and of itself.

Finally, to test whether compound function in zebrafish was conserved in mammals, we tested whether treatment of rat primary oligodendrocytes with splitomicin also affected the mammalian oligodendrocyte lineage. Indeed, we found that two-day treatment of cultures immediately following plate-down (see Materials and methods) resulted in a robust increase in the number of MBP expressing oligodendrocytes (*Figure 9*). Furthermore, we assessed proliferation (using an antibody recognising Ki67) of oligodendrocyte lineage cells (using an antibody recognising olig2). We saw an increase in the number of Ki67-expressing, olig2-expressing cells, upon splitomicin treatment (*Figure 9*), indicating that the observed increase in myelinating oligodendrocyte number was due to increased proliferation and subsequent differentiation of a larger pool of OPCs. Therefore, our identification of splitomicin as a regulator of the oligodendrocyte lineage in zebrafish was validated as having the capacity to regulate the mammalian oligodendrocyte lineage. Future studies will investigate the conservation of function of our other hits in the mammalian oligodendrocyte lineage.

## Discussion

Here we describe the development of a fully automated, sub-cellular resolution, in vivo chemical screening pipeline, which we have used to identify compounds that promote myelinating oligodendrocyte formation in zebrafish. By combining the now commercially available LP Sampler and VAST-BioImager (both Union Biometrica) with a spinning disk confocal microscope (VAST-SDCM), we automated the delivery, orientation, and high-speed, sub-cellular resolution imaging of zebrafish. We developed an image handling and analysis pipeline that automates file handling tasks, and which also allows assay-specific operations including region of interest selection and a variety of quantitative assessments, such as analyses of cell number, distribution, shape and fluorescence intensity. We have implemented and exemplify the power of this pipeline to identify compounds that regulate myelinating oligodendrocyte lineage formation. Importantly, our imaging platform and image analysis scripts can in principle be adapted to execute a variety of assays relevant to numerous biological processes of interest to the community. Our custom scripts are freely available at https://github.com/jasonjearly/VAST-

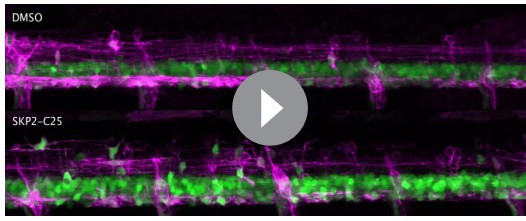

**Video 6.** Time-lapse movies of Tg(Olig2:GFP), Tg (sox10:mRFP) double transgenic control (top) and SKP-C25-treated (bottom) embryos imaged between 58 hpf and 70 hpf. Refers to *Figure 4—figure supplement 1*.
https://elifesciences.org/articles/35136#video6

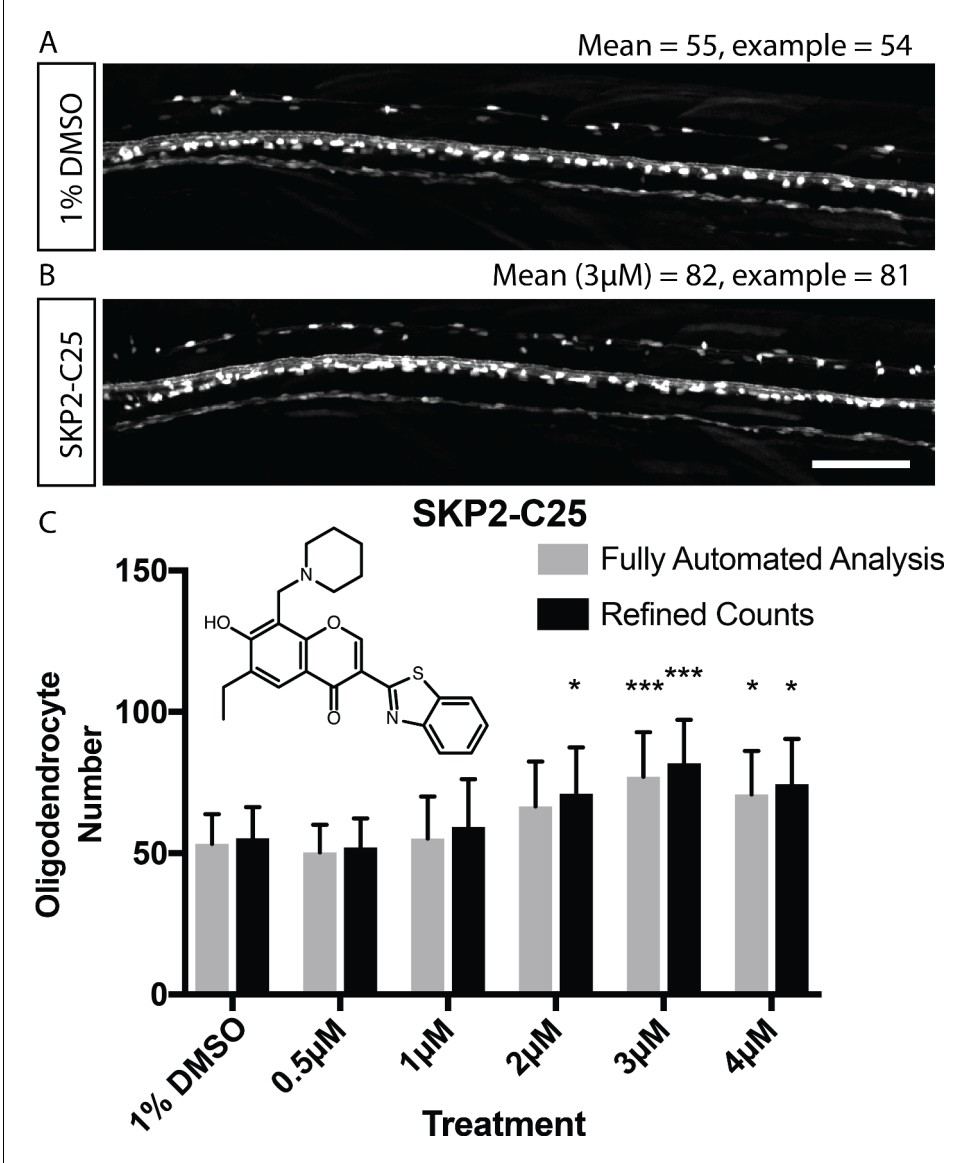

**Figure 4.** Validation of automated image analysis to identify changes in myelinating cell number induced by chemical hit. (**A**, **B**) Confocal images of Tg(mbp:EGFP) control (**A**) and SKP2-C25-treated (**B**) animals reflecting mean cell number per treatment (numbers refer to entire dorsal spinal cord, not magnified area of same animals). Scale bar = 100 μm. (**C**) Quantitation of cell number in control and SKP2-C25 treated animals using fully automated (grey bars) and refined counts, auto ROIs (black bars). Two-way ANOVA followed by Dunnett's multiple comparison test was used to assess statistical significance, with multiple comparison adjusted P value presented as *p<0.05, ***p<0.001. Error bars represent means ± s.d.

The online version of this article includes the following figure supplement(s) for figure 4:

**Figure supplement 1.** SKP2-C25 increases proliferation of OPCs in the dorsal spinal cord.

SDCM (**Early, 2018**; copy archived at https://github.com/elifesciences-publications/VAST-SDCM).

## Scaling up and refining high-resolution quantitative in vivo screens

Here we demonstrate the power of the VAST-SDCM platform and automated image handling and analysis scripts by carrying out a small screen of compounds, which were tested for their ability to promote oligodendrocyte lineage progression. We identified three novel compounds that reproducibly increased myelinating oligodendrocyte number (C646, GSK-J5 and splitomicin). However, each

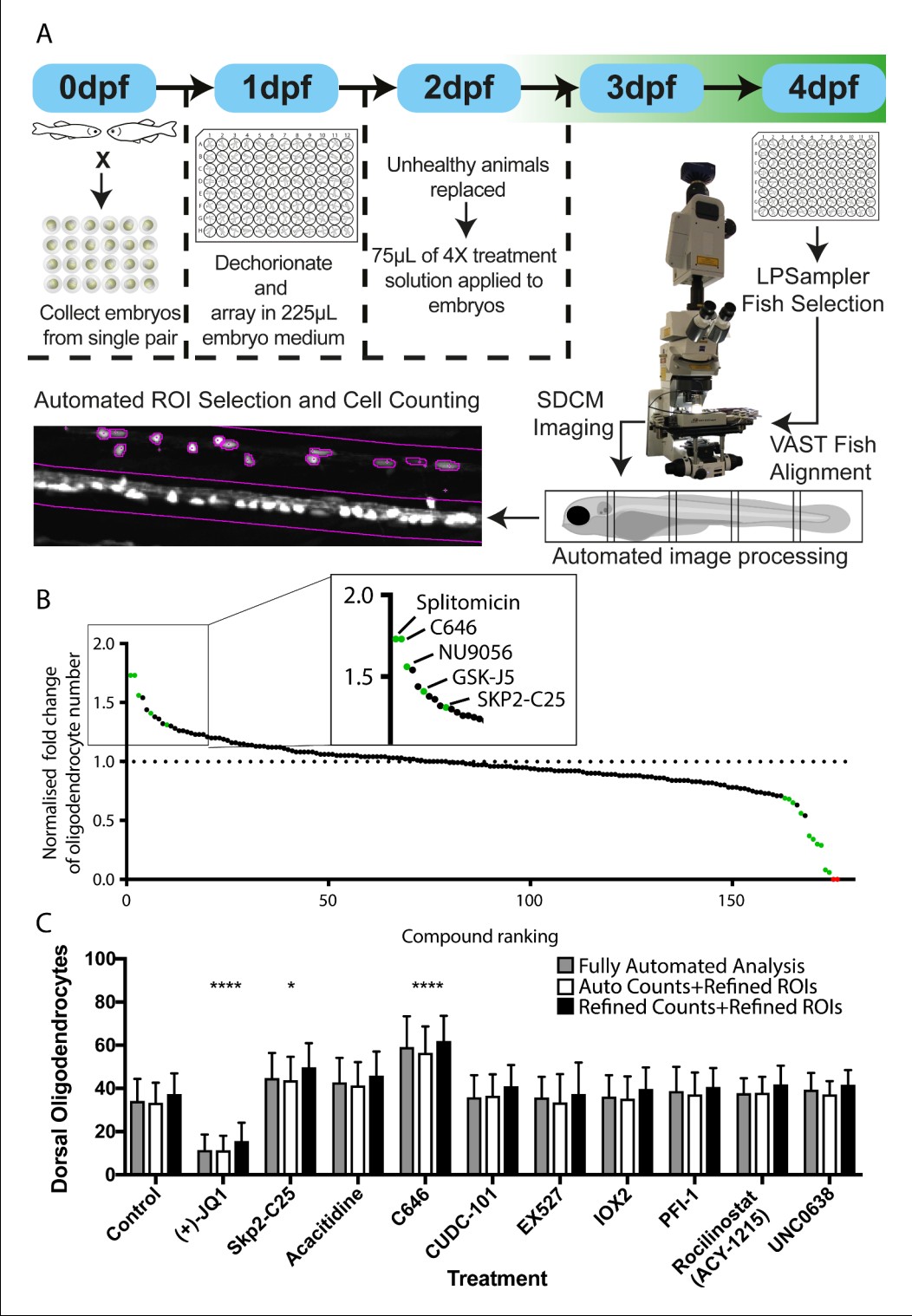

**Figure 5.** Automated chemical screening pipeline in zebrafish identifies novel regulators of oligodendrocyte development. (**A**) Schematic of chemical screening pipeline. (**B**) Results of primary screen displayed as normalised fold change in oligodendrocyte number relative to matched DMSO-treated controls and ranked by cell increase for all 183 treatments (146 unique compounds) with hit compounds that increase or decrease cell number marked in green and toxic compounds in red. Inset shows ranking of compounds that significantly increased oligodendrocyte cell number. See also **Table 2** and **Supplementary file 1** (**C**) Quantification of dorsal

*Figure 5 continued on next page*

*Figure 5 continued*

oligodendrocyte number from a single screening session showing comparison of different analysis methods.
*p<0.05, ****p<0.0001. Error bars represent means ± s.d.
The online version of this article includes the following figure supplement(s) for figure 5:

**Figure supplement 1.** Example hits identified from our primary screen as regulators of oligodendrocyte development.

---

of these compounds, and our positive control compound, SKP2-C25, functioned over a relatively tight concentration range, and exhibited somewhat variable effects. Indeed, SKP2-C25 only served its function as a positive control one out of the two times that it was tested in the screen, and C646 only exhibited statistically significant increases in oligodendrocyte number in one of the three times it was screened, although it did have trend increases of similar effect size all three times (see *Supplementary file 1* and *2*). Interestingly, the durations of screening runs in which these compounds were not returned as hits were longer than those from which they were, reflecting the fact that variability in cell number will increase over time, given that continuously developing live animals are being examined. The false negatives may also be due to the narrow concentration windows of efficacy (*Figures 4* and *6*), coupled with the fact that researchers typically don't assess actual compound concentration prior to a screen. These examples serve to highlight the issue of false negatives that need to be addressed with any chemical screening platform. To reduce variability due to protracted screening runs, we recommend screening individual compounds over shorter periods of time, when using live animals, by adjusting their distribution across screening plates accordingly. In addition, to mitigate false negatives due to restricted windows of efficacy, one could carry out high-resolution screens at multiple concentrations within doses well tolerated by the animal. This would clearly require the processing of more animals per screen and represent a trade-off between increased throughput and reduced error rate. We have demonstrated that for our oligodendrocyte cell number assay, imaging and screening a single tile instead of the entire animal would have been as effective in hit identification, and more than halved the screening time. This would have allowed us to assess about 85 animals per hour, instead of 40. To further scale up screening capacity on this platform, another option may involve pooling distinct compounds, and assessing the ability of pools to elicit phenotypes. One could then subsequently identify the effective compound(s) in the active

---

**Table 2.** Significant Hits From Screen.
Fold change in oligodendrocyte number induced by individual hit compounds normalised to DMSO treated controls in appropriate treatment plate. For complete list of compound effects and putative targets, see *Supplementary file 1*.

| Compound | Fold change (Treated/Control) | P value |
|---|---|---|
| Splitomicin | 1.73 | 0.0001 |
| C646 | 1.73 | 0.0001 |
| NU9056 | 1.56 | 0.0128 |
| GSK-J5 (hydrochloride) | 1.41 | 0.0031 |
| SKP2-C25 | 1.31 | 0.0404 |
| Zebularine | 0.69 | 0.0498 |
| Apicidin | 0.68 | 0.0139 |
| Suberohydroxamic Acid | 0.65 | 0.0043 |
| IOX1 | 0.56 | 0.0003 |
| CUDC-907 | 0.37 | 0.009 |
| PI3K/HDAC Inhibitor I | 0.30 | 0.0006 |
| (+)-JQ1 | 0.29 | 0.0001 |
| JIB-04 (NSC693627) | 0.08 | 0.0001 |
| Trichostatin A | 0.06 | 0.0001 |

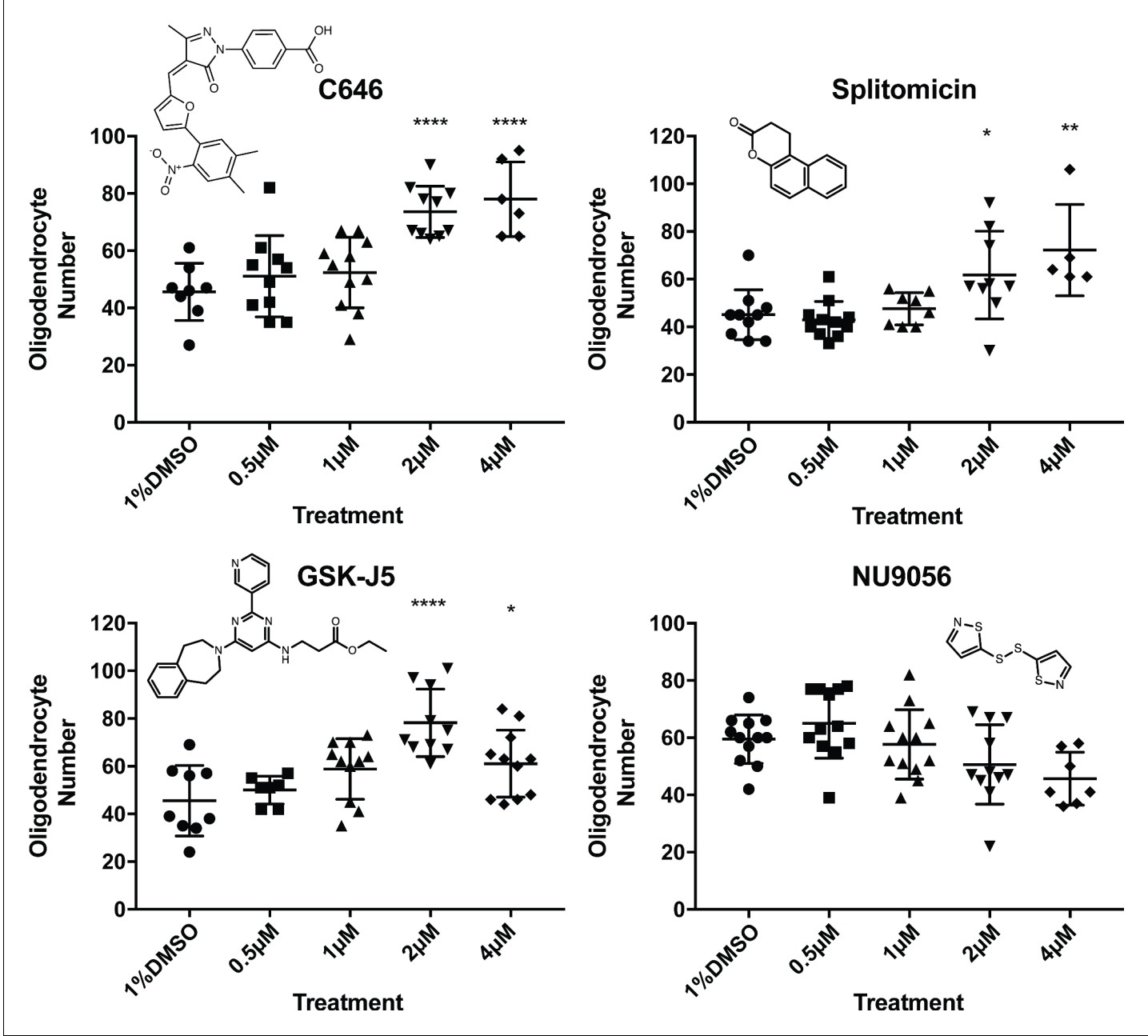

**Figure 6.** Concentration series and chemical structures of hit compounds from screen. For each hit identified in the primary screen, repeat treatments were carried out as before (12 fish treated from 2-4dpf), with either 1% DMSO, 0.5 µM, 1 µM, 2 µM, 4 µM, 10 µM or 20 µM of the hit treatment. For all compounds, the 10 and 20 µM treatments were fatal and are not shown on the above graphs. Chemical structures of each hit are displayed in respective graphs. One-way ANOVA followed by Dunnett's multiple comparison test was used to assess statistical significance, with multiple comparison adjusted P value presented as *p<0.05, **p<0.01, ****p<0.0001. Error bars represent means ±s.d.

pool(s). The pooling of compounds is most likely to be effective for compounds with limited toxicity and thus will differ library to library. However, this may be an effective way in which to greatly scale up the utility of the VAST-SDCM system.

To best ensure reliability of screens and to reduce error rates in so far as possible, the use of positive controls and appropriate powering of experiments is essential. We would advise investigators initiating quantitative screens that aim to identify subtle phenotypes to run robust positive controls throughout individual screening runs where possible. Prior to the initiation of our screen, our power calculations indicated that a reduced n = 6 would have had a power of 80% to detect changes in cell

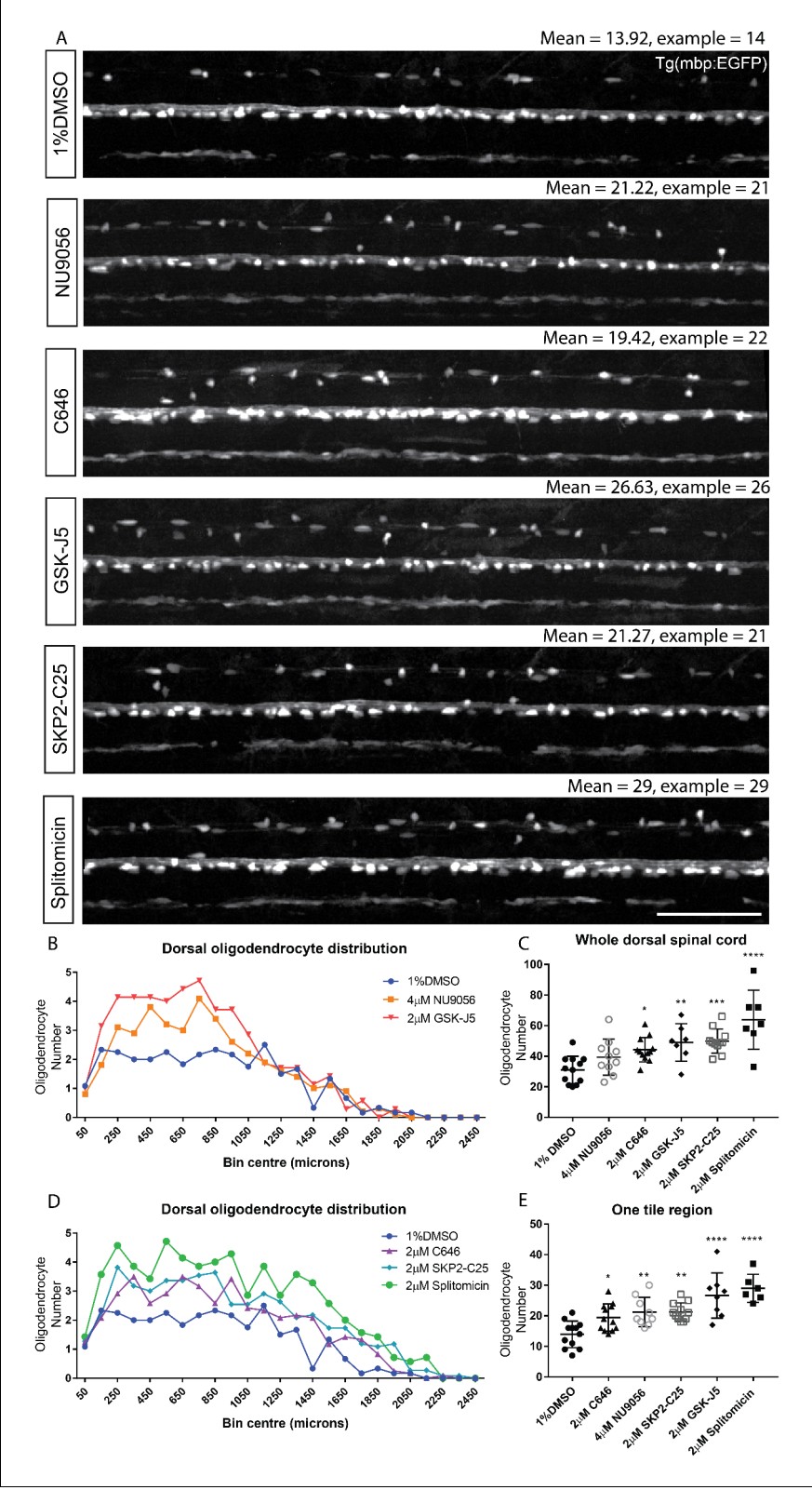

**Figure 7.** Automated region-specific analyses of identified hit compounds. (**A**) Representative single tile images of 4 dpf Tg(mbp:EGFP) larvae following treatment with the five hit compounds that increased oligodendrocyte number in our screen. Scale bar, 100 µm. Mean numbers of dorsal oligodendrocytes quantified for each treatment and in the examples shown are indicated above each corresponding image. (**B**) Plot of the anterior-posterior

*Figure 7 continued on next page*

*Figure 7 continued*

distribution of dorsal oligodendrocytes along the spinal cord demonstrates region-specific effects with NU9056 and GSK-J5 treatment, whereas C646, SKP2-C25 and splitomicin exhibit more consistent increases in cell number along the A-P axis (D). (C) Quantification of oligodendrocyte number throughout the entire dorsal region shows lower sensitivity for identifying compounds with region-specific effects compared with quantification within one tile region (second of five tiles) (E). *p<0.05, **p<0.01, ***p<0.001, ****p<0.0001. Error bars represent means ± s.d.

number of 35% (vs 95% for n = 9, which we scaled to 12 to account for occasional failure to image samples correctly). Indeed, the success rate of fish loading through to automated data analysis was 80% across the entire screen, meaning that on average our actual n was just over nine per condition. Following our screen, we analysed our raw data and assessed what hits we would have identified, if we compared our actual n for each hit compound compared to n = 6 or n = 3 (*Supplementary file 4*). Interestingly, at an n = 6 only C646 and SKP2-C25 would have been identified as hits, and the absence of identifying the validated hits splitomicin and GSK-J5 would have represented false negatives. In addition, at the lower n values, we would have identified one false positive hit at n = 6 and two at n = 3 (*Supplementary file 4*), and incurred six false negatives at n = 6 and 12 at n = 3 (*Supplementary file 4*).

Our desire to test the ability of the system to identify hits that regulated cell number by 35% was stringent. If a larger effect size threshold was used, then a lower 'n' would suffice. For example, if we defined a criterion for a hit as having the ability to change myelinating cell number by >75% with 95% power, then in principle an n of 3 would suffice. Therefore, in the design of any quantitative screen, one needs to consider the definition of what constitutes a hit and to power the analysis based on known variability in the parameter being assessed accordingly.

## Novel regulators of oligodendrocyte formation

Following our screen, we identified and validated four compounds that promote myelinating oligodendrocyte formation in larval zebrafish. Splitomicin is a small molecule inhibitor of the sirtuin family of NAD$^+$-dependent HDACs (*Bedalov et al., 2001*), with reported activity for sirtuin 1 and 2 (sirt1 and sirt2) in mammalian cells in vitro (*Pasco et al., 2010*). Sirt1 shows broad protein expression throughout the rodent CNS (*Sidorova-Darmos et al., 2014*), but has been implicated in regulating the oligodendrocyte lineage. For example, conditional genetic inactivation of sirt1 in adult rodent neural stem cells led to an increase in OPC formation, and a consequent increase in remyelination following toxin-induced demyelination (*Rafalski et al., 2013*). Interestingly, the opposite effect on the oligodendrocyte lineage has been reported in neonatal animals, with reduced numbers of proliferating OPCs following sirt1 knockdown and increased oligodendrocyte differentiation under hypoxic conditions suggesting a context-dependent role for sirt1 (*Jablonska et al., 2016*). In contrast to sirt1, sirt2 expression is highly enriched in the oligodendrocyte lineage, and is upregulated in newly differentiating and myelinating oligodendrocytes (*Werner et al., 2007*; *Zhang et al., 2014*). However, despite its specific expression in the oligodendrocyte lineage, the role of sirt2 in regulating oligodendrocyte differentiation and myelination remains somewhat unclear (*Ji et al., 2011*; *Li et al., 2007*). Our data suggest that splitomicin increases myelinating oligodendrocyte number in both zebrafish and rat, but it remains unclear whether these effects are mediated via sirt1 and/or sirt2 or indeed on another unknown target. Therefore, the validation of targets engaged by hit compounds, by genetic and biochemical means, is essential to disentangling the mechanistic basis of compound function. In addition, future genetic investigation of the role of prospective targets in a cell-type specific manner will be required to explore their relative roles in the oligodendrocyte lineage and in other cell types. Although we did not see general effects on animal development with our hit compounds, we cannot rule out roles in other cells, particularly as these regulators are broadly expressed. Our second ranked hit, C646, is a competitive inhibitor of the CBP/p300 family of histone acetyltransferases (HATs) (*Bowers et al., 2010*). As transcriptional co-activators, CBP and p300 have been implicated in coordinating cell fate decisions through their interactions with numerous histone and non-histone related proteins in vitro (*Nakashima et al., 1999*; *Zhang et al., 2016*). Notably, CBP can acetylate the transcription factor olig1 and regulate its nuclear to cytoplasmic localisation, which is associated with oligodendrocyte differentiation (*Dai et al., 2015*), and p300 levels have been shown to be altered in a subset of patients with MS (*Pedre et al., 2011*). The third novel hit

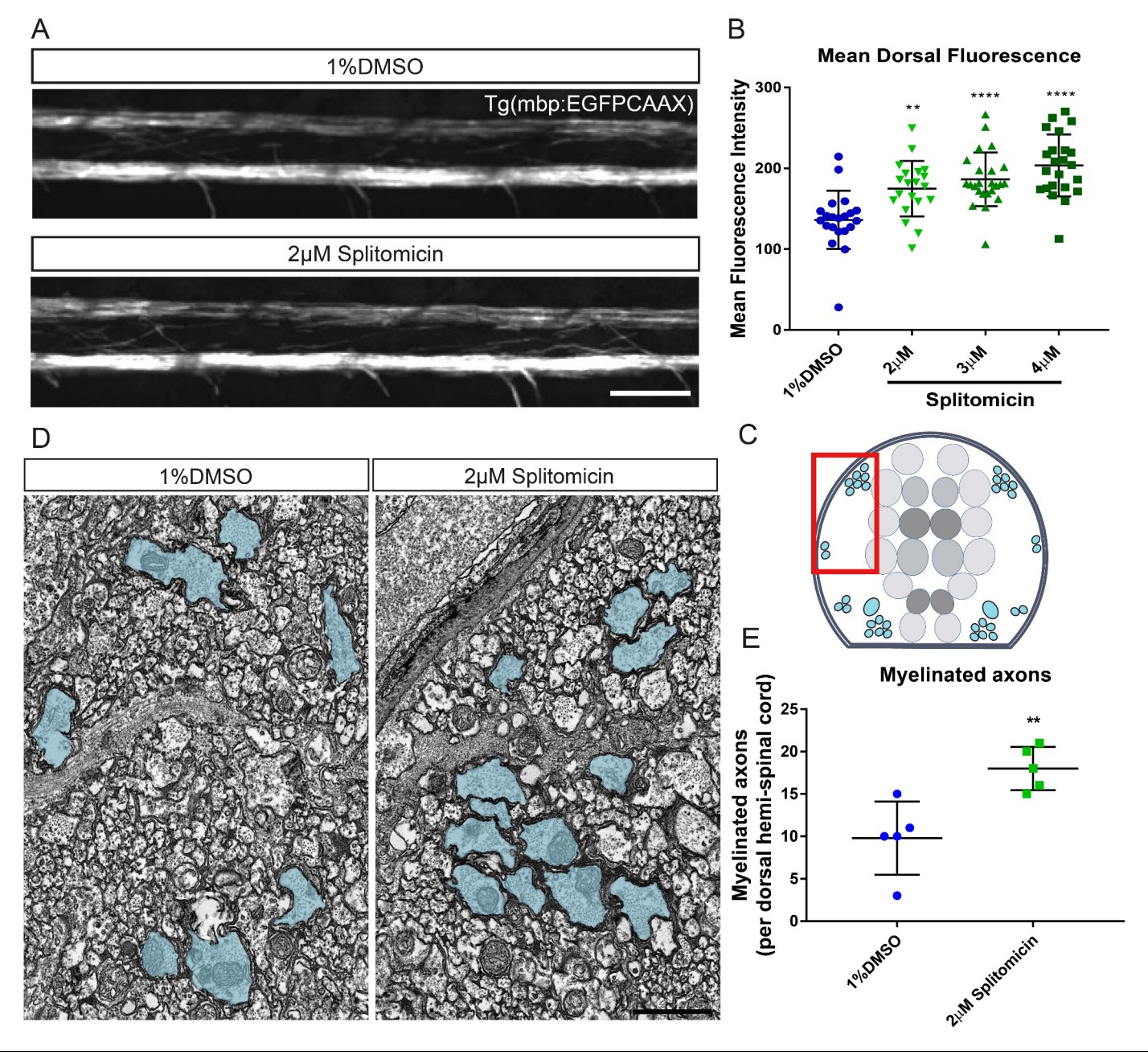

**Figure 8.** Treatment with splitomicin increases spinal cord myelination in vivo. (**A**) Representative images of myelin within the spinal cord of control (top) and splitomicin-treated (bottom) Tg(mbp:EGFP-CAAX) larvae. Scale bar, 50 µm (**B**) Quantification of mean fluorescence intensity throughout the entire dorsal spinal cord demonstrates a significant increase following 2–4 µM splitomicin treatment from 2-4 dpf. (**C**) Schematic of a transverse section through the zebrafish spinal cord shows the position of myelinated axons (blue) in relation to the gray matter (neuronal cell bodies are coloured in grey). Red box indicates approximate area shown in (**D**) in which myelinated axon number was quantified. (**D**): Transmission electron micrographs of control (left) and splitomicin-treated (right) larvae at 4 dpf with myelinated axons pseudo-coloured in blue. Scale bar, 1 µm (**E**) Numbers of myelinated axons in the dorsal spinal cord are significantly increased following 2–4 dpf treatment with 2 µM splitomicin. **p<0.01, ****p<0.0001. Error bars represent means ± s.d.

The online version of this article includes the following figure supplement(s) for figure 8:

**Figure supplement 1.** Automated fluorescent quantification of myelin during development.

that was validated in all subsequent assays, GSK-J5, displays weak activity for the jumonji (JMJ)

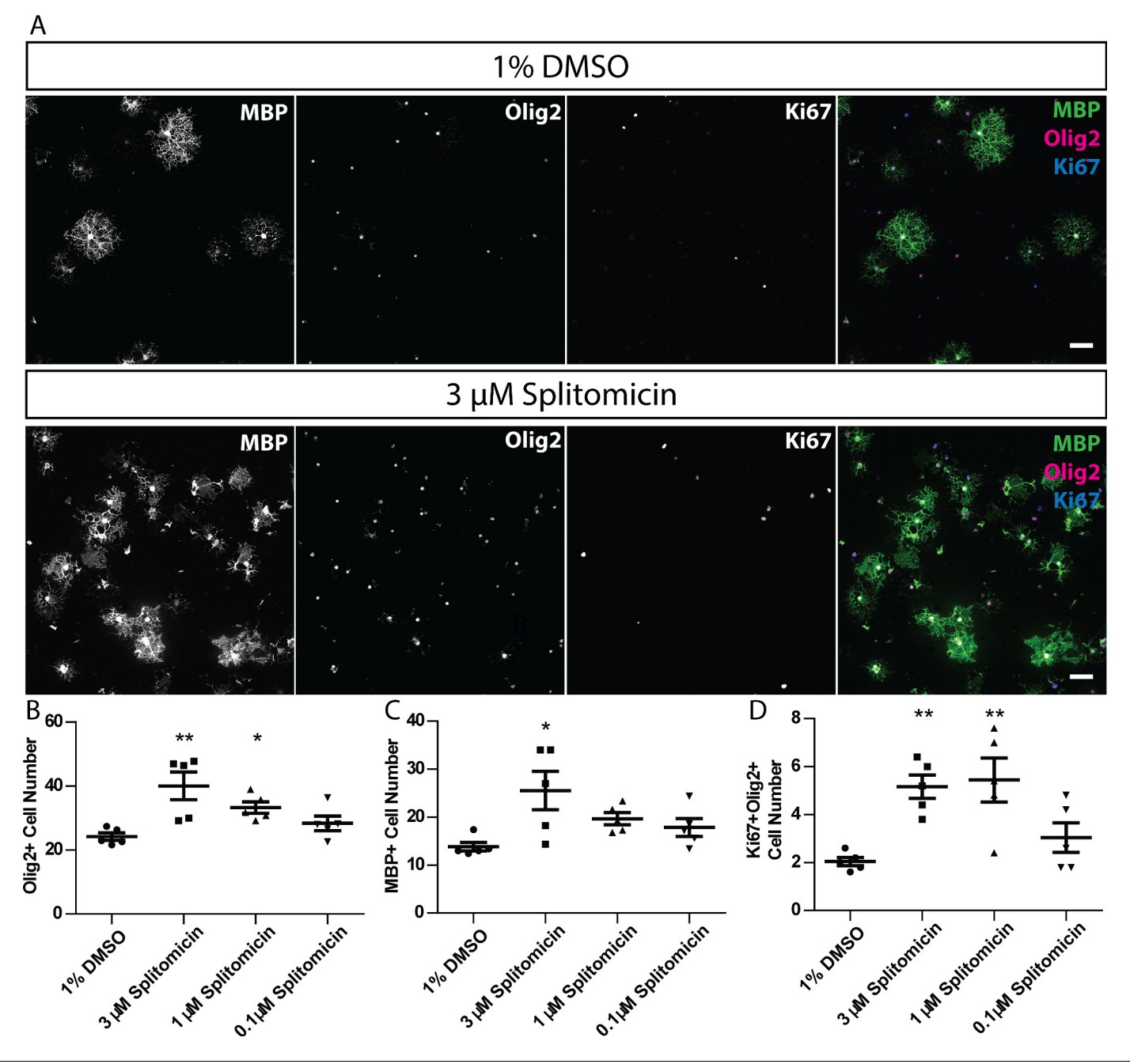

**Figure 9.** Splitomicin enhances mammalian oligodendrocyte generation in vitro. (**A**) Representative images of rat oligodendrocyte cultures treated with 1% DMSO or 3 µM splitomicin, stained for antibodies that recognise MBP, Olig2 and Ki67. Scale bar = 50 µm. (**B–D**) Quantification of Olig2+ cell number (**B**), MBP+ cell number (**C**) and Ki67+ Olig2+ cell number (**D**). Shapiro-Wilk normality test determined that datasets were not normally distributed. Therefore, a Kruskal-Wallis test was followed by Dunn's multiple comparison test to test statistical significance between conditions. *p<0.05, **p<0.01. Error bars represent means ± s.e.m.

family of histone demethylases in mammalian in vitro assays and for this reason is recommended as a structure-matched inactive control for the more active isomer, GSK-J4 (*Kruidenier et al., 2012*). This fact, coupled with the finding that GSK-J4 and other selective JMJ inhibitors within the library had no effect on oligodendrocyte number in our screen, suggests that GSK-J5 may promote myelinating oligodendrocyte formation through an alternative target, again highlighting the issue of target validation (*Schenone et al., 2013*). From the point of view of both hit and target validation, it is

important to note the value of using multiple compounds (ideally structurally diverse) that are purported to hit the same target. Given that most compounds are likely to bind to more than one target in vivo, if several compounds with an overlapping target elicit the same phenotype, then one has a better foundation upon which to pursue the more challenging genetic and biochemical methods of target validation.

Our pilot screen identified hits that robustly increase myelinating oligodendrocyte number. However, extensive further studies will be required to fully characterise the mechanisms by which these compounds affect specific stages of the oligodendrocyte lineage. The ability to carry out time-lapse imaging of oligodendrocyte lineage progression in vivo using zebrafish offers a great advantage in assessing when compounds mediate their functions, as shown for SKP2-C25 (*Figure 4—figure supplement 1* and *Video 6*). Future time-lapse studies will explore how our other hit compounds affect the dynamic development of the oligodendrocyte lineage in vivo. In addition, the development of further stage-specific reporters and markers in zebrafish coupled with complementary studies in cell based platforms allow further dissection of the timing and cellular mechanisms of compound activities.

## Versatility and future potential of the VAST-SDCM platform

We exemplify the potential of the VAST-SDCM system for discovery screens using a transgenic reporter of myelinating oligodendrocytes, and myelination, but essentially any reporter of interest can be imaged using this platform. The system has multiple excitation laser lines and two cameras that allow simultaneous imaging in two channels without any increase in imaging time, meaning that two assays could be screened in parallel, given appropriate reporters, or several reporters in parallel with some increase in imaging times. In addition, the platform can accommodate photo-ablation and photo-activation lasers for acute manipulations, and the stability of larvae in the imaging capillary, and speed of the SDCM imaging system is such that up to minutes long time-lapse microscopy could, in principle, be carried out on individual animals. A recent update to the liquid handling system now means that the platform can be configured such that zebrafish can be dispensed back in to the wells of multi-well plates from the BioImager, allowing for collection of embryos for further study, for example genotyping or re-imaging.

In addition to the obvious versatility of the imaging platform, we have also extended our image analysis scripts to provide additional functionalities. One common issue when dealing with fluorescently labelled reporters or markers is expression in tissues outwith areas of interest, or disruptive auto-fluorescence, for example in the zebrafish skin. To exemplify this utility, we segmented the spinal cord from its surrounding peripheral tissue in Tg(mbp:nls-EGFP) animals, which also express nuclear-localised EGFP in peripheral Schwann cells (*Figure 2—figure supplement 3*, *Video 4*, and see Materials and methods). In principle, our landmark-based segmentation approach can be implemented in a case by case reporter-dependent manner for selection of 3D ROIs. However, in situations where there are no opportunities to assign a reference landmark, one could assign ROIs by determining average distances from the reference landmark in our transgenic reporters. We executed our screen using 2D images, because we found that cell number could be as accurately assessed from 2D MIPs as 3D z-stacks. However, in some cases information on the 3D nature of manipulations will be of importance. Therefore, we added another optional functionality to our image analysis pipeline, which allowed us to encode quantitative information relating to individual images in a z-stack in 2D MIP images (*Figure 2—figure supplement 3* and see Materials and methods). This approach has the benefit of allowing one to assess effects of experimental manipulations in 3D without the storage and processing issues associated with dealing with high-content 3D imaging datasets. In general, it is desirable in a screen to maximise data use and minimise costs (monetary and time) associated with storage and processing of very large datasets.

In addition to serving as a powerful discovery system, zebrafish present numerous advantages for the deep phenotypic characterisation of compound function on biological processes. In particular, their suitability for live imaging and the myriad transgenic reporters of various molecules, cells and tissues, allow the effects of compounds on cell behaviours to be followed with high spatial and temporal resolution in the living animal in ways that are hard to achieve in other vertebrates (*Bin and Lyons, 2016*). Furthermore, the ability to directly observe tissue development and health at scale and over time-scales of days, presents remarkable advantages for drug development projects that need to triage chemical structures with possible deleterious effects. In addition to serving as a

platform for phenotypic discovery screens, the VAST-SDCM is also suited to more focussed target-driven studies, for example the profiling of arrays of compounds designed to hit individual targets. Coupled with the capacity to image essentially any molecule, cell, or tissue type with a sufficiently robust fluorescent reporter, the VAST-SDCM platform provides a versatile option for scalable (medium–throughput) high-resolution focussed approaches. Automated higher-throughput, lower resolution in vivo screening capacities have recently also been demonstrated using zebrafish (*Wang et al., 2015*). The availability of screening options at multiple scales highlights the power of zebrafish as a discovery model. We expect that our implementation of VAST-SDCM and our adaptable image handling and analysis scripts will be of value to the wider research community, and we invite interested users to contact us.

## Materials and methods

### Zebrafish husbandry and transgenic lines

Adult zebrafish were housed and maintained in accordance with standard procedures in the Queen's Medical Research Institute zebrafish facility. All experiments were performed in compliance with the UK Home Office, according to its regulations under project licenses 60/4035 and 70/8436. Adult zebrafish were subject to a 14/10 hr, light/dark cycle. Embryos were produced by pairwise matings and raised at 28.5°C in 10 mM HEPES-buffered E3 Embryo medium or conditioned aquarium water with methylene blue. Embryos were staged according to hours or days post-fertilisation (hpf or dpf). The following transgenic lines were used for this study: Tg(mbp:EGFP) (*Almeida et al., 2011*), Tg (mbp:EGFP-CAAX) (*Almeida et al., 2011*), Tg(mbp:nls-EGFP) (*Karttunen et al., 2017*), Tg(olig2: EGFP) (*Shin et al., 2003*) and Tg(sox10:mRFP) (*Kirby et al., 2006*).

### Hardware setup (VAST-SDCM platform)

We used the Large Particle (LP) Sampler and VAST BioImager (Union Biometrica) to automate the delivery of zebrafish from the wells of multi-well plates to the BioImager microscope platform. To combine the power of the VAST BioImager platform to automatically orient zebrafish within a thin-walled glass capillary according to user-defined templates with high-speed imaging, we mounted the BioImager platform on an upright Axio Examiner D1 (Carl Zeiss) microscope, fitted with a high-speed CSU-X1 spinning disk confocal scanner (Yokogawa, Tokyo, Japan). All animals were imaged from a lateral view. In order to optimise the field of view, we added a 1.6X post-magnification adaptor to the light path, which when combined with a C-Plan-Apochromat 10X (0.5NA) dipping lens (Carl Zeiss) and dual AxioCam 506 m CCD cameras (Carl Zeiss), allowed acquisition of images with X-Y dimensions of 662 µm x 492 µm, such that the majority of the capillary's 600 µm Y-axis could be captured. The imaging system was configured with 50 mW 405 nm, 50 mW 488 nm and 20 mW 561 nm lasers and two AxioCam 506 m CCD cameras, high-speed emission filter wheel with 450/50, 520/35 and 527/54 + 645/60 nm emission filters for camera 1, and a fixed 630/98 nm emission filter for simultaneous acquisition.

To ensure rapid image acquisition in the z-plane, we employed a long-range, high-speed PIFOC P-725.4CD piezo objective scanner (Physik Instrumente, Karlsruhe, Germany), capable of up to 400 µm high speed travel.

The VAST BioImager microscope platform can be moved in the x, but not y-axis. To generate stitched z-stacks the system needs to hand control back and forth (via Transistor-Transistor Logic, TTL, commands, *Figure 1*) between the VAST BioImager software (Union Biometrica) and the microscope's image acquisition software, Zen Blue 2.0 (Carl Zeiss). This allows interspersed rounds of acquisition of individual z-stacks and movement of the BioImager platform in the x-axis (*Figure 1*). For higher resolution imaging, a C-Plan Apochromat 20X (1NA) dipping lens (Carl Zeiss) was employed.

### Custom software

All custom software and scripts used in this work are freely available at https://github.com/jason-jearly/VAST-SDCM (*Early, 2018*).

## Automation of file handling

All file handling was automated using custom ImageJ Macro Language (IJM) scripts (*Schindelin et al., 2012*; *2015*). Due to the nature of the communication between the VAST BioImager and microscope (a simple TTL signal), the images acquired by the microscope contain no fish or well-specific information. To overcome this issue, our image processing scripts include functions to cross reference image acquisition times with the VAST BioImager signal times (recorded by VAST in a .csv file) for specific tile locations for individual fish for each well sampled. This data was then used to concatenate upper and lower z-stacks, stitch tiles of z-stacks, generate maximum intensity projections of tiled z-stacks, and finally to perform cell counts. File stitching was carried out using FIJI's inbuilt Grid/Collection Stitching plugin (*Preibisch et al., 2009*). Additional quality control checks were applied to the output of the stitching plugin to ensure that stitched image dimensions fell within a given tolerance of those expected. Any images outwith these dimensions are highlighted to the user. This file management process is carried out by the Process_VAST_v1.1.ijm macro (*Early, 2018*).

## Automation of ROI selection

ROI selection was performed using the plotROI function of our custom image analysis script, Count-Cells_2D_v1.1.ijm (*Early, 2018*). This function first identifies the ventral spinal cord of the Tg(mbp: EGFP) transgenic zebrafish line by repeated sampling of 150 µm blocks of the image along the x-axis, with the sum of grey values for each y-position within a block used to identify the peak fluorescence in y for the block (see *Figure 2*). This process was carried out using the getProfile function of ImageJ for a line of width 150 µm. The stereotyped nature of the fluorescence in this transgenic line means that the identified peaks delineates the ventral spinal cord of the embryo. Due to the sparse distribution of cells in the posterior spinal cord, an additional step was added to find the final cell. A best fit straight line was produced from the last four successful ventral spinal cord points and extrapolated caudally. This line was sampled to identify the last potential cell falling within 20 µm, above or below this best fit (*Figure 2*).

Using the line representing the ventral spinal cord of the fish, regions of interest were then set for the dorsal and ventral spinal cord, as well as the entire structure. Polygon regions are created with characteristic dimensions. In addition, in order to mimic the narrowing of the spinal cord, the dorsal region was tapered linearly towards half its original height, at the posterior (*Figure 2*). All regions of interest were then saved for subsequent inspection or repeated measurements.

Whilst this method has been used to segment the dorsal spinal cord using the Tg(mbp:EGFP) reporter, the principle of this technique could be broadly applied to other structures of interest.

## Automated cell counting

Cell counts were carried out using the CountCells_2D_v1.1.ijm macro (*Early, 2018*). Cell counting was first performed within the regions identified by the plotROI function, using ImageJ's findMaxima function. The findMaxima function requires an input threshold which sets the minimum difference in fluorescence between two potential maxima to allow them to be identified as distinct peaks (i.e. the minimum drop in fluorescence between two peaks).

Whilst initial tests showed that the findMaxima function was able to identify differences between positive and negative control treatments, at lower input threshold values, many false positive points were produced (data not shown). At these lower thresholds, brighter cells possess multiple maxima, due to lack of fluorescence homogeneity (green arrows in *Figure 3*), while at even lower thresholds, background noise may be identified as a cell.

To prevent over assignment of false positive hits using the findMaxima function, the processCells function was developed to refine the raw findMaxima point selection. For each point identified by the findMaxima function, ImageJ's doWand function was used to select connected pixels (with a tolerance of half that point's peak intensity). Statistics were then produced for the putative cell using the getStatistics and ROI.getBounds functions of ImageJ. The size and shape of the produced selection were then assessed for cell likeness, and false positives removed. Additional cell likeness criteria (such as minimum, maximum or standard deviation of fluorescence intensity) can readily be applied by minor modification to the processCells function. Maxima which did not fit the cell likeness criteria were discarded, and refined point as well as polygon outline selections were saved for further

analysis (see binned cell counts method, below). Appropriate thresholds were identified for a given dataset using the intelligentThresh function. This function takes user adjusted cell counts from five randomly selected fish and identifies the threshold which gave the closest match to the human cell count.

## Binned cell counts

The relative distribution of cells along the length of the spinal cord was automatically determined using a custom ImageJ script (see script Plot_Distribution_v1.1.ijm, *Early, 2018*). This script takes the point selection and the ventral spinal cord selection produced by the cell counting script, for each fish, and assigns an x-coordinate to each point. This x-coordinate was calculated by identifying the closest point on the ventral spinal cord to each cell. This calculation was necessary to account for the variable curvature of the spinal cord as well as any minor fish positioning errors. Cell numbers were then binned based on these calculated x-positions.

## Tg(mbp:EGFP-CAAX) intensity measurements

Relative quantities of myelin were estimated between treatments using ImageJ's 'getRawStatistics' function, which measured the mean grey value within the whole zebrafish spinal cord, as selected by the plotROI function, with manual user adjustment of the ventral spinal cord selection to ensure an accurate selection. The measurement was performed on stitched maximum intensity z-projections for each fish, using an adapted version of the CountCells_2D macro, see CAAX_Analysis_v1.1.ijm (*Early, 2018*).

## 3D volume segmentation

A custom ImageJ script (see script 3D_Crop_v1.1.ijm, *Early, 2018*) was written to present a user with orthogonal maximum intensity projections of a z-stack, to which selections may be made and added to ImageJ's ROI manager. The script identifies ROIs wholly contained within the xy, xz or yz regions, and sets all pixels outside these regions to 0 for all slices, in their respective view. This creates an AND effect, where only pixels included in all ROIs are retained. The script outputs a new z-stack with the retained pixels and, optionally, the discarded pixels.

## Depth projection

To efficiently map the depth of maximum intensity for each pixel in xy of acquired z-stacks, the inbuilt ImageJ ZProjector plugin was adapted (see D_Projector_v1.1.jar, *Early, 2018*) to record this information in a secondary 2D image (termed the depth projection, or dProj). This secondary dProj image was then masked with the cell profiles created by the processCells function, and any non-cell pixel set to 0 for the purposes of illustration (*Figure 2—figure supplement 3D*). A lookup table was applied to the subsequent image to illustrate the additional contrast possible with this method.

## Sample size calculation

A sample size calculation was performed using the method described recently by (*White et al., 2016*). The following equation was used to calculate the required sample size (n), based on a sample data set of DMSO treated and positive control treated animals (data not shown).

$$ n = \frac{2\sigma^2 \left(Z_\beta + \frac{Z_\alpha}{2}\right)^2}{\left(\mu_\rho - \mu_\eta\right)^2} \times 1.15 $$

Where $\sigma$, $Z_\beta$, $Z_\alpha$, $\mu_\rho$ and $\mu_\eta$ are standard deviation (of control with largest variance), required power, required level of statistical significance, mean of positive control group and the mean of negative control group respectively.

## Chemical screen: larvae preparation and compound treatments

Out-crosses of the Tg(mbp:EGFP) transgenic line were used for the primary screen. At 24 hpf, embryos were enzymatically dechorionated using protease from Streptomyces griseus (0.5 mg/mL for 6 min) (Sigma-Aldrich, St. Louis, MO) and manually arrayed into 96-well plates in a volume of 225 µL E3 media (three embryos per well). At 48 hpf, larvae were checked for viability, and any unhealthy

animals were manually removed and replaced in equal volumes with clutch-matched larvae. 10 mM compound stocks in DMSO (Xcess Biosciences, San Diego, CA; SelleckChem, Houston, TX; Cayman-Chem, Ann Arbor, MI) were serially diluted using a multi-channel pipette to a 4X concentrated treatment solution. Between 48–51 hpf, 75 µL of this 4X stock was added directly into the larval wells for a final concentration of 2 µM in 1% DMSO. Each compound was tested on 4 wells of 3 fish per well (n = 12) and compared against 1% DMSO negative controls within the same plate. Treatment plates were then kept within secondary chambers to reduce evaporation and incubated under standard temperature conditions for 2 days without compound refreshment. At four dpf, larvae were anaesthetised with 600 µM tricaine prior to imaging. For the preliminary qualitative screen, embryos were manually mounted in 1.1% low-melting agarose (Invitrogen) on a glass coverslip, and suspended over a microscope slide using high vacuum silicone grease to create a well containing embryo medium and 600 µM tricaine.

Treatments were imaged such that replicate wells were imaged throughout the day (i.e. Treatment 1 well 1, Treatment 2 well 1, Treatment 3 well 1..., Control well 1, **then** Treatment 1 well 2, Treatment 2 well 2, Treatment 3 well 2..., Control well 2...), to spread any variability due to development whilst imaging live animals over time.

## Time-lapse imaging

For time-lapse imaging Tg(sox10:mRFP; olig2:GFP) larvae were used. Zebrafish larvae were treated at 48 hpf with SKP2-C25 (1% DMSO). At 56–57 hpf, larvae were anaesthetised in 600 µM tricaine and manually mounted in 1.5% low melting point agarose in embryo medium containing either SKP2-C25 (1% DMSO) or 1% DMSO and 600 µM tricaine. Time-lapse imaging was performed using a Zeiss LSM710 confocal microscope (Carl Zeiss) with a temperature controlled stage set to 28.5°C throughout imaging. Images of the middle region of the spinal cord (somite 15 = centre of image) were acquired using a 20X objective (Zeiss Plan-Apochromat 20X dry, NA = 0.8), zoom 1, corresponding to a 425 µm region of the spinal cord. Dual colour Z-stacks (488 and 594 nm lasers) were acquired every 14 min over 16 hr of the full depth of the spinal cord with 2 µm Z-interval. Larvae were checked the following morning for viability and general health with any unhealthy animals excluded from analysis. For analysis, maximum-intensity projections were made for each time point over the 16 hr imaging period. To reduce x-y drift, dual-colour images were registered using the olig2:EGFP channel and 'Register virtual stack slices' plugin in Fiji. The 'Transform virtual stack slices' plugin was then used to apply the same transformations to the sox10:mRFP channel. Dorsal OPC migration and proliferation events were manually quantified over a 12 hr period from 58 to 70 hpf using the Cell Counter plugin in Fiji.

## In vitro OPC differentiation assay

Cortical mixed glial cultures were generated from Sprague Dawley rats P0-2, as described previously (*McCarthy and de Vellis, 1980*). After 10–12 days of expansion oligodendrocyte progenitor cells were mechanically separated, counted and grown on poly-D-lysine coated 18 mm coverslips at a density of 75,000 cells per well. Cells were grown in myelination media: 50:50 DMEM:Neurobasal Media, B27 supplement, 5 µg/mL N-acetyl cysteine, and 10 ng/mL D-biotin, ITS and modified Sato (100 µg/mL BSA fraction V, 60 ng/ml Progesterone, 16 µg/ml Putrecsine, 400 ng/mL Tri-iodothyroxine, 400 ng/mL L-Thyroxine), at 37°C in 7.5% CO2. Treatments were added at plating and refreshed after 24 hr.

Cells were fixed after 48 hr with 4% PFA for 15 min and blocked for 30 min in 10% goat serum and 0.1% triton-X-100 at room temperature. Primary antibodies were diluted in blocking solution and applied overnight at 4°C: mouse anti-Olig2 (1:200, Millipore; MABN50), rat anti-MBP (1:250; AbD Serotec; MCA409S) and rabbit anti-Ki67 (1:250; Abcam; ab16667). Cells were incubated with fluorescently conjugated secondary antibodies diluted in blocking solution (1:1000, Life Technologies-Molecular Probes), for 1 hr at room temperature and counterstained with Hoechst (5 µg/ml). Five random images were taken per coverslip on a Leica SP8 confocal microscope (20X objective) and counted using ImageJ, double blind to condition. A Shapiro-Wilk normality test was used to determine normality of distribution. Because data were not normally distributed, a Kruskal-Wallis test was followed by Dunn's multiple comparison test to test statistical significance between conditions. *p<0.05, **p<0.01, ***p<0.001, ****p<0.0001. Error bars represent means ± s.e.m.

## Transmission electron microscopy

Tissue was processed for electron microscopy as previously described (*Czopka and Lyons, 2011*). Briefly, 4 dpf larvae were anaesthetised with tricaine and chemically fixed with microwave stimulation using a solution of 2% glutaraldehyde, 4% paraformaldehyde in 0.1 M sodium cacodylate buffer (Agar Scientific, Essex, UK). Embryos subsequently underwent secondary fixation with microwave stimulation in 2% osmium tetroxide in 0.1 M sodium cacodylate/imidazole (Agar Scientific) followed by en bloc stain in saturated uranyl acetate (~8%, w/v, in water) (TAAB Laboratories Equipment Ltd., Berkshire, UK). Samples were then progressively dehydrated in a series of ethanol and acetone washes before being embedded in Embed-812 resin (Electron Microscopy Sciences, Hatfield, PA). Ultra-thin sections (70 nm thickness) were cut around somite 15 using a Reichert Jung Ultracut microtome (Leica Microsystems, Wetzlar, Germany) and stained using uranyl acetate (TAAB Laboratories Equipment Ltd.) and Sato lead stain (see (*Czopka and Lyons, 2011*) for detailed protocol). Stained sections were imaged using a Jeol JEM-1400 Plus transmission electron microscope (JEOL USA, Inc., Peabody, MA) at 8600x magnification. Individual tiles were automatically aligned using the Photo-Merge tool in Adobe Photoshop CC (Adobe Systems Inc., San Jose, CA) and quantified manually using Fiji (ImageJ).

## Statistical analyses

All data are expressed as mean ± SD, unless otherwise stated. Statistical tests were carried out using GraphPad Prism 7. All automated analysis was performed by investigators blind to the experimental treatment. For identification of hits from our primary screen, statistical significance was determined by one-way ANOVA, with Dunnett's multiple comparisons test performed on treatments within individual experiments (from single clutches) compared to 1% DMSO controls. All other statistical analysis was performed using one-way or two-way ANOVA with Dunnett's or Tukey's multiple comparisons test where applicable or by two-tailed Student's t-test as stated in the text. See *Supplementary file 3* for details of raw data and tests per individual experiments and their relevant figure panel.

## Acknowledgements

We would like to thank members of the Lyons group for helpful comments on the manuscript, and the University of Edinburgh zebrafish facility staff for support. This work was supported by a Wellcome Trust Senior Research Fellowship (102836/Z/13/Z) and a Lister Institute Research Prize to DAL. The implementation of VAST-SDCM was supported through a BBSRC ALERT award to DAL and Catherina Becker and E Elizabeth Patton (University of Edinburgh), and by contributions from the College of Medicine and Veterinary Medicine, University of Edinburgh, and Biogen.

## Additional information

### Competing interests

Katy LH Marshall-Phelps, Jill M Williamson: funded by a collaborative grant from Biogen for part of the period of this project. Hari Kamadurai, Marc Muskavitch: was employed by Biogen at the time of the study. The other authors declare that no competing interests exist.

### Funding

| Funder | Grant reference number | Author |
|---|---|---|
| Wellcome | 102836/Z/13/Z | David A Lyons |
| Lister Institute of Preventive Medicine | | David A Lyons |
| Biogen | | David A Lyons |

The funders had no role in study design, data collection and interpretation, or the decision to submit the work for publication.

## Author contributions
Jason J Early, Conceptualization, Software, Formal analysis, Validation, Investigation, Visualization, Methodology, Writing—original draft, Writing—review and editing; Katy LH Marshall-Phelps, Conceptualization, Data curation, Formal analysis, Validation, Investigation, Visualization, Methodology, Writing—original draft, Writing—review and editing; Jill M Williamson, Investigation, Writing—review and editing; Matthew Swire, Formal analysis, Investigation; Hari Kamadurai, Marc Muskavitch, Resources, Writing—review and editing; David A Lyons, Conceptualization, Supervision, Funding acquisition, Investigation, Visualization, Writing—original draft, Project administration, Writing—review and editing

## Author ORCIDs
Jason J Early [iD] https://orcid.org/0000-0003-4313-6445
Katy LH Marshall-Phelps [iD] https://orcid.org/0000-0001-6275-5941
David A Lyons [iD] https://orcid.org/0000-0003-1166-4454

## Ethics
Animal experimentation: All animal studies were carried out with approval from the UK Home Office and according to its regulations, under project licenses 60/8436 and 70/8436. The project was approved by the University of Edinburgh Institutional Animal Care and Use Committee.

## Decision letter and Author response
Decision letter https://doi.org/10.7554/eLife.35136.sa1
Author response https://doi.org/10.7554/eLife.35136.sa2

# Additional files

## Supplementary files
• Supplementary file 1. Spreadsheet documenting the effect of all screened compounds, with their putative targets, on myelinating oligodendrocyte number, arranged by effect size. Mean cell number effect size is normalized to DMSO-treated sibling controls, per screening session. Dunnett's multiple comparison test was used to assess statistical significance, with multiple comparison adjusted P value presented.

• Supplementary file 2. Detailed spreadsheet documenting the effect of all screened compounds, with their putative targets, and raw data of compound effect on myelinating oligodendrocyte number, n numbers per compound screened, standard deviations and statistical analyses. As per *Supplementary file 1*, mean cell number effect size is normalized to DMSO-treated sibling controls, per screening session. Dunnett's multiple comparison test was used to assess statistical significance, with multiple comparison adjusted P value presented.

• Supplementary file 3. Overview of statistical tests carried out as per corresponding figure panel.

• Supplementary file 4. Overview of statistical significance for Dunnett's multiple comparison test with P values adjusted for multiple comparisons when data from *Supplementary file 1* is analysed with all samples, the first 6 or 3 samples from each treatment. Results are only shown for treatments where at least one test was found to be significant. *p<0.05, **p<0.01, ***p<0.001, ****p<0.0001 and ns = not significant (p>=0.05).

• Transparent reporting form

## Data availability
All data generated or analysed during this study are included in the manuscript and supporting files.

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
