## [Decision Letter]

Thank you for submitting your article "An automated high-resolution in vivo screen to identify chemical regulators of myelination" for consideration by *eLife*. Your article has been reviewed by three reviewers, and the evaluation has been overseen by Tanya Whitfield as the Reviewing Editor and Didier Stainier as the Senior Editor, The following individual involved in review of your submission has agreed to reveal their identity: Jason Rihel (Reviewer #3).

The reviewers have discussed the reviews with one another and the Reviewing Editor has drafted this decision to help you prepare a revised submission. In the discussion among the editors and reviewers, we feel your contribution would be appropriate to consider in the category of a Tools and Resources paper. The standards for such a contribution are just as high as for a Research Article, but the criteria for acceptance hinge on the technical excellence and utility of the work and a bit less on the mechanistic insight generated by the paper.

Summary:

Early et al., describe the development of an automated screening pipeline to identify chemical regulators of myelination in zebrafish. They have used mbp:GFP transgenic zebrafish lines to mark the myelinating glial cells and the VAST embryo liquid handler to orientate each fish precisely so that images of the CNS can be processed and quantified in an identical way. Using this platform, they have screened a library of 175 epigenetic targeted compounds and assayed for an increase in myelinating oligodendrocyte number. Five hit compounds were found, one of which they indicate is able to increase myelination in vivo.

Essential revisions:

As you will see, while reviewers 1 and 3 are largely positive, reviewer 2 has more substantial concerns. After discussion between the reviewers, it was agreed that some of these should be addressed. Essential revisions are:

1) Please address the concerns from reviewer 2 over reproducibility, false positives and false negatives. Comparison with previously validated compounds will be important here.

2) Please provide more information about the use of positive controls and show clearly that a positive control compound acts as expected in the screen.

3) If a suitable mutant line is available, please address the suggestion from reviewer 1 to test the ability of hit compounds to rescue myelination in a fish with reduced myelination.

4) Please provide more information about the structures of the hit compounds, as suggested by reviewer 1.

5) Some further validation of the compounds is required, to give more insight into action of the compounds and their effects on oligodendrocyte biology. The reviewers have made several suggestions. After discussion, it was agreed that validation in a mammalian system would be beyond the scope of this manuscript, and so this is not essential. However, further validation in the zebrafish model is required.

Reviewer #1:

The strength of this paper is in the detailed description of the assay and the automated image analysis. The data is of a high quality with clear and beautifully presented figures. The method they describe is novel and timely and will be of great interest to other researchers in the field. Orientating zebrafish larvae for high through-put image analysis has always been challenging, making power scores difficult to determine; using the VAST and image processing method described here will enable more robust screening assays to be undertaken in vivo. The down side to this is the time taken to screen a 96 well plate of fish and the number of compounds that can be screened lowers the through-put.

The automated pipeline for ROI determination and subsequent image analysis works well and is comparable to manual determination. This should be adaptable to many other projects. This screen identified five compounds that increased oligodendrocyte number. However, as HDACs are already known to increase oligodendrocyte number, the fact the library screened identified HAT inhibitors was not surprising, but, this provides a good validation of this approach.

The further characterisation of the compounds was limited and did not include an analysis of a dose response for all five compounds, and only a limited dose range for their hit compound splitomicin was tested. They did analyse if myelin levels increased with splitomicin too and found they could detect this using their automated analysis, however it would have been interesting to see if this compound could also rescue myelin in a myelin deficient fish.

Reviewer #2:

This manuscript details work setting up a high-throughput in vivo screen of oligodendrocyte formation in the zebrafish embryo and testing this screening pipeline using a library of 175 compounds targeting enzymes involved in epigenetic and post-translational modifications. By using an in vivo system, the authors can assess toxicity or other global effects of compounds that would preclude their utility in drug development.

Any successful screening pipeline should several major features: a) High-throughput capacity b) Reproducibility c) Limit false positives and false negatives

The authors have successfully established a high-throughput method for assessing the formation of oligodendrocytes in zebrafish embryos. By automating the majority of the process, they are able to expedite data acquisition and analysis and eliminate human bias. Importantly, their automated processing is comparable to their manual processing (Figure 3D, Figure 4C).

However, this screening pipeline has serious shortcomings in features b (reproducibility) and c (limiting false positives and negatives). Furthermore, the lack of follow-up on the hits in this screen results in a manuscript that does not provide substantially new information on oligodendrocyte biology.

1) Reproducibility. Within the screen itself (looking at Figure 2—figure supplement 2), 3 of the 5 hits that increase oligodendrocyte formation do not replicate this increase. The other 2 compounds are not tested more than once within the screen. For example, in experiment 1, the authors test SKP2-C25 (n=12; 1.31 fold increase, p = 0.0404). When the authors test this compound again in experiment 2, there is no increase in oligodendrocyte formation (1.08 fold increase, p=0.9634) with n=22 and a lower standard deviation for both the control and the SKP2-C25 condition than in experiment 1, indicating that this lack of statistical difference in experiment 2 is not due experiment 2 being underpowered as compared to experiment 1. NU9056 does not replicate the statistically significant increase from experiment 3 in experiment 9, and also does not replicate using the same methods in Figure 5C. C646 does replicate the statistical significant finding in experiment 1 in experiments 9 and 10.

This lack of reproducibility within the screen casts doubt on the ability of the pipeline to draw correct conclusions with high levels of confidence based on any one experiment – substantially limiting the value of this screening pipeline.

It is important to highlight this data outside of the supplemental tables, so that readers can take this information into consideration when evaluating this system.

2) False positives and false negatives. Since at least 3 compounds were found to both significantly increase and not change oligodendrocyte formation, this pipeline has returned either false positives or false negatives – both of which should be minimized in a screen.

Is the screening platform able to detect real hits consistently? The authors should test previously validated compounds that increase oligodendrocyte formation in this screening platform to verify that these compounds show a consistent increase in oligodendrocyte formation.

Are these hits false positives? The authors should provide further validation of these hits – particularly those that both increased and had no effect on oligodendrocyte formation depending on the experiment – outside of this screening platform. Since the goal is to translate these compounds to treat human diseases, assessing the effects in a mammalian system would be a highly relevant way to further validate these hits.

3) Why do various compounds targeting common molecules in similar manners (i.e. inhibiting sirtuin 1 and 2) have profoundly different effects on oligodendrocyte formation in this screen?

4) A distinct disadvantage to this screening protocol as compared to others that have been previously published (e.g. Deshmukh et al., 2013; Mei et al., 2014) is that there is no method for discerning how each compound is affecting oligodendrocyte formation (e.g. directly acting on OPCs to cause an increase in OPC differentiation). Different diseases disrupt different aspects of oligodendrocyte formation and elucidating on which aspect of oligodendrocyte formation a compound is acting is important for discovering compounds useful for distinct diseases. As it stands, this manuscript does not provide substantial insights into oligodendrocyte biology. How do the identified compounds alter oligodendrocyte lineage cell biology? Visualizing both OPCs and oligodendrocytes could be a method by which the authors could provide some additional information on the mechanism of action of a compound. Which cell type(s) do they act on?

Reviewer #3:

Overall this is a well-presented manuscript with well-documented results. The major advance of this study is mainly in the technical and coding work to make the VAST Bioimager, the spinning disk, and the analysis pipeline smoothly talk to each other to automatically pull out segmented, high resolution images for automated cell counting. This is a highly useful set of code in the zebrafish screening world, as it allows for automated analysis of specific image features, even in transgenic lines that may label other cell types, allowing one to even find interesting regionalized effects, for example in sensitivities across the A-P axis. There are other technical papers on the screening capabilities of the VAST Bioimager and related imaging methodologies, but this paper does still feel like an addition to these.

By fishy standards, the throughput of the screening example of only a few hundred compounds is low - only 175 compounds at a single dose vs. typical zebrafish screens these days of 1000s or even 10K compounds, but this is reasonable proof of principle for this kind of high-resolution imaging screen. The example of myelination is a well-chosen one to show off the capabilities, as such a screen would otherwise be a challenge. What might have been nice is a John Henry vs. the locomotive kind of comparison - i.e. how quickly could automated counters do the same screen, and what kinds of screening throughputs might now be possible and over what timescales? This might have really made the power of this approach clear.

Another weakness is that the manuscript is quite light on the biology- the compound hits haven't provided a clear biological insight as yet, nor even a clear target for further study. But I understand that *eLife* publishes technical/methods/resource papers and this would be a good example of that.

I don't have any major specific concerns. The code is well documented and freely available, the screening data is thorough, and the pipeline is well described.

[Editors' note: further revisions were requested prior to acceptance, as described below.]

Thank you for resubmitting your work entitled "An automated high-resolution in vivo screen in zebrafish to identify chemical regulators of myelination" for further consideration at *eLife*. Your revised article has been favorably reviewed by Didier Stainier (Senior Editor), Tanya Whitfield (Reviewing Editor), and three reviewers.

As you will see, all three reviewers agree that the manuscript has been improved. However, there are some remaining comments from reviewers 1 and 2 that need to be addressed before acceptance. It should be possible to address these comments by simple edits to the manuscript, and so we hope that they will be quick and straightforward to complete. Please see the reviews below for details:

Reviewer #1:

Early et al., have made substantial changes to the text and figures of their manuscript to address the comments from all the reviewers. These include new experiments to address the validity of the hits identified in the screen and reworking of the text. Overall these changes have improved the manuscript and clarified areas that raised concern. Although some of these revisions have also highlighted other issues about the reproducibility of the results, the text has been modified to make it suitable for a Tools and Resources publication and I agree this is a better fit for the manuscript. The detailed methodology and analysis presented here will be of interest to anyone attempting small molecule screens and I therefore think the manuscript is now suitable for publication. I thank the authors for the modifications they have made and for responding to each of my points, including the re-evaluation of the power number data. I have made a few specific comments on the authors responses below and included a few very minor changes to the text.

1) The authors have used a dose response assay on their top four hits and validated three of the compounds. One of these hits NU9056 was no longer able to increase cell number, although it did still show a localized effect in a different assay. The authors attribute the variability seen between different experiments in part to the short window of action for the compounds thus highlights the importance of having good controls and finding optimum doses and this is now discussed in the text. I think one of the issues here is that the control compound is still being validated and isn't optimal in all experiments and therefore is effectively an extra hit compound. The authors do discuss this issue in their response. I also wonder if the variability seen is linked to the order of the compounds when being imaged and the time taken. The authors have discussed varying the order of the compounds imaged on the VAST in response to a point made previously and it would be of interest to know if there is any correlation.

2) To further characterize the action of the hit compounds the authors have used live imaging to show the control compound has increased proliferation. As this is quite a generic effect, it would be worth to note if this is specific to the OPCs or if there was an increase in all olig2:GFP and sox10:mRFP cells.

One of the hits, Splitomicin was also shown to increase MBP positive cells in rat oligodendrocyte cultures, further validating this hit compound in a different species and a different experimental assay as requested by one of the reviewers and this has improved the manuscript.

As there are just three strong hits it would be of interest to know if these two different approaches (live imaging, rat oligodendrocytes) had been tested with the other compounds (and control). While I think that having the data for one compound validates the approach and is fine for this manuscript, it might be worth mentioning if these experiments with the other compounds are planned or if they were inconclusive.

Reviewer #2:

The authors have addressed most of the previous comments, however, there are a few remaining points to be addressed:

1) Numerous times throughout the text, the authors state that SKP2-C25 and splitomicin promote OL lineage progression into OLs through increasing OPC proliferation (new data). Could the authors please elaborate on their suggestion that promoting OPC proliferation (where there are minimal asymmetric cell divisions that generate OLs (Hughes et al., 2013) could cause an increase in OL numbers? Known pro-proliferative compounds such as PDGF-AA are well-documented to decrease OL formation in vitro. In any case, proliferation does not necessarily promote lineage progression.

2) In response to reviewer comments, the authors have increased the clarity of their data by including a discussion of the inconsistent reproducibility and false positives/negatives within the screen. This is important data - especially when considering the novel screening platform. In the spirit of transparency, the authors should report false negatives in the Results section. It is appropriate to report the non-significant trials of C646 and NU9056 when reporting their significant trials in detail (subsection “Chemical screen for compounds that regulate myelinating oligodendrocyte number”, C646; NU9056). The non-significant trial of SKP2-C25 could be reported here or elsewhere in the Results section.

3) Data from Figure 6 does not seem to be in Supplementary file 3. Additionally, please comment in the text or legend on how this data was obtained. Was it using the fully automated screening platform or another method?

Reviewer #3:

This resubmission has substantial new data, including evidence that the candidate molecules found in the fish screen also work on mammalian cells in vitro. They have rewritten the manuscript to highlight more clearly the automation methods and detail several screening issues that can be addressed in various ways.

The new submission has addressed all of my concerns.

---

## [Author Response]

The reviewers have discussed the reviews with one another and the Reviewing Editor has drafted this decision to help you prepare a revised submission. In the discussion among the editors and reviewers, we feel your contribution would be appropriate to consider in the category of a Tools and Resources paper. The standards for such a contribution are just as high as for a Research Article, but the criteria for acceptance hinge on the technical excellence and utility of the work and a bit less on the mechanistic insight generated by the paper.

We are very grateful for the opportunity to submit a revised version of our manuscript. We agree that the manuscript would be well suited to the Tools and Resources category and resubmit it accordingly. We have added a significant amount of new data and data analysis to our manuscript to address feedback and have reoriented much of our Discussion section to fit better with the Tools and Resources format.

The main experimental and data analysis additions are:

In-depth characterisation of the effects of SKP2-C25 on the oligodendrocyte lineage. As we clarify in the manuscript, this compound was the positive control that we used in our screen. We now provide dose response information on the compound effect on oligodendrocyte number (new Figure 4). We also provide time-lapse imaging based data showing that this compound promotes OPC proliferation in the developing zebrafish (new Figure 4—figure supplement 1 and Video 6).

We also provide dose response studies for the four additional compounds that increase oligodendrocyte number, which addresses many of the points brought up regarding reproducibility. These data are now presented in a new Figure 6.

Also relating to the issue of screen refinement and reproducibility, we have carried out a post hoc analysis of our screen to test how effective different “n” numbers would have been in identifying validated hits. This analysis is now presented as a Supplementary file 4.

We also show data that indicate that our top hit compound, splitomicin, increases myelinating oligodendrocyte number in rat oligodendrocyte cultures, validating conservation of effect across species.

We believe that these additions greatly strengthen our manuscript and are grateful to the reviewers for helping us produce a better paper.

[…] Essential revisions:As you will see, while reviewers 1 and 3 are largely positive, reviewer 2 has more substantial concerns. After discussion between the reviewers, it was agreed that some of these should be addressed. Essential revisions are:1) Please address the concerns from reviewer 2 over reproducibility, false positives and false negatives. Comparison with previously validated compounds will be important here.

We have addressed these important points in detail in our revision, by the inclusion of a significant amount of new data, data analysis and discussion.

Point by point revisions are outlined throughout the original reviews.

The principal new data of relevance to the issues of reproducibility, false positives and negatives are concentration series tests of hit compounds. We show that 3 of the 4 hit compounds identified as increasing oligodendrocyte number were validated through concentration series analyses. However, all compounds exerted their functions over relatively narrow concentration ranges, which likely contributes to the variability seen in the screen. These data are now presented in a new Figure 6.

One of our hit compounds did not show activity in increasing oligodendrocyte number in this concentration series analysis but did exhibit a region-specific effect in increasing oligodendrocyte number in an independent experiment (Figure 7). Therefore, this is not considered a fully validated hit, and would not be pursued further.

To further test how robust our screen was in identifying the validated hits, we carried out a new post hoc analysis to test how many hit compounds would have been identified with differing “n” values. We found that the n number we selected was indeed the most robust, given that a significant number of false positives and negatives were apparent in taking lower “n” numbers. These data are now presented in a new Supplementary file 4.

We also now discuss in much greater detail the issue of false positives and negatives in screening and outline important refinements that should be considered for future quantitative screens.

2) Please provide more information about the use of positive controls and show clearly that a positive control compound acts as expected in the screen.

We now clarify that the compound SKP2-C25 was used as the positive control in designing our screen, and that it acted as expected in our screen. We also discuss the fact that this compound occasionally exhibited variable effects, due to a narrow effective concentration range (shown in new Figure 4) as well as solubility issues and was therefore not an optimal control. However, we did not have an alternative positive control for our assay prior to initiating our screen. We have now identified compounds that more robustly increase oligodendrocyte number and we discuss the need for robust positive controls in executing quantitative screens and for applying rigorous criteria for defining hits and powering screens accordingly.

3) If a suitable mutant line is available, please address the suggestion from reviewer 1 to test the ability of hit compounds to rescue myelination in a fish with reduced myelination.

Given the prospective targets of the compounds identified in our screen, there is not a suitable mutant phenotype to try to rescue with these compounds. However, instead, to strengthen our manuscript and further validate the ability of our assay to identify compounds of relevance to myelination, we added an experiment which tested how our hit compound splitomicin affected mammalian oligodendrocyte lineage progression. These new experiments showed that splitomicin also increased myelinating oligodendrocyte number in rat in vitro cultures and did so by promoting oligodendrocyte progenitor cell proliferation. These data are now shown in a new Figure 9.

4) Please provide more information about the structures of the hit compounds, as suggested by reviewer 1.

We have added this information as requested, as part of a new figure in which we carry out assessments of how hit compounds affect myelinating oligodendrocyte number across a range of concentrations. These data are shown in new Figure 4 and Figure 6.

5) Some further validation of the compounds is required, to give more insight into action of the compounds and their effects on oligodendrocyte biology. The reviewers have made several suggestions. After discussion, it was agreed that validation in a mammalian system would be beyond the scope of this manuscript, and so this is not essential. However, further validation in the zebrafish model is required.

In addition, we also found that splitomicin increased rat oligodendrogenesis in vitro (new Figure 9).

Reviewer #1:[…] The automated pipeline for ROI determination and subsequent image analysis works well and is comparable to manual determination. This should be adaptable to many other projects. This screen identified five compounds that increased oligodendrocyte number. However, as HDACs are already known to increase oligodendrocyte number, the fact the library screened identified HAT inhibitors was not surprising, but, this provides a good validation of this approach.The further characterisation of the compounds was limited and did not include an analysis of a dose response for all five compounds, and only a limited dose range for their hit compound splitomicin was tested. They did analyse if myelin levels increased with splitomicin too and found they could detect this using their automated analysis, however it would have been interesting to see if this compound could also rescue myelin in a myelin deficient fish.

We thank the reviewer for their feedback, and now provide dose response data for all five compounds that increase myelinating oligodendrocyte number. These data are now shown in new Figure 4 and Figure 6. Given what is known about the prospective targets of our hit compounds, a suitable mutant was not available to test for possible rescue. However, to further characterize the effects of our compounds, we now include time-lapse imaging data showing how treatment with SKP2-C25, promotes OPC proliferation in vivo. These data are shown in a new Figure 4—figure supplement 1 and Video 6. In addition, we tested how splitomicin affected development of rat oligodendrocytes in vitro, validating our findings in zebrafish. These data are now shown in a new Figure 9.

Reviewer #2:[…] 1) Reproducibility. Within the screen itself (looking at Figure 2—figure supplements 2), 3 of the 5 hits that increase oligodendrocyte formation do not replicate this increase. The other 2 compounds are not tested more than once within the screen. For example, in experiment 1, the authors test SKP2-C25 (n=12; 1.31 fold increase, p = 0.0404). When the authors test this compound again in experiment 2, there is no increase in oligodendrocyte formation (1.08 fold increase, p=0.9634) with n=22 and a lower standard deviation for both the control and the SKP2-C25 condition than in experiment 1, indicating that this lack of statistical difference in experiment 2 is not due experiment 2 being underpowered as compared to experiment 1. NU9056 does not replicate the statistically significant increase from experiment 3 in experiment 9, and also does not replicate using the same methods in Figure 5C. C646 does replicate the statistical significant finding in experiment 1 in experiments 9 and 10.This lack of reproducibility within the screen casts doubt on the ability of the pipeline to draw correct conclusions with high levels of confidence based on any one experiment – substantially limiting the value of this screening pipeline.It is important to highlight this data outside of the supplemental tables, so that readers can take this information into consideration when evaluating this system.2) False positives and false negatives. Since at least 3 compounds were found to both significantly increase and not change oligodendrocyte formation, this pipeline has returned either false positives or false negatives – both of which should be minimized in a screen.Is the screening platform able to detect real hits consistently? The authors should test previously validated compounds that increase oligodendrocyte formation in this screening platform to verify that these compounds show a consistent increase in oligodendrocyte formation.Are these hits false positives? The authors should provide further validation of these hits – particularly those that both increased and had no effect on oligodendrocyte formation depending on the experiment – outside of this screening platform. Since the goal is to translate these compounds to treat human diseases, assessing the effects in a mammalian system would be a highly relevant way to further validate these hits.

We thank the reviewer for highlighting their concerns around reproducibility, false positives and false negatives in our screen. We entirely agree with the reviewer that we should have addressed these points in detail in our original manuscript. We now add substantive new data that validates 4/5 compounds that increase oligodendrocyte number, and which do indeed identify one compound as a marginal hit that would not warrant further investigation.

We provide dose response data on all five compounds that increase oligodendrocyte number and find that 4 of these compounds exhibit phenotypes in these studies, albeit over a relatively narrow concentration window (see new Figure 4 and Figure 6). Indeed, we speculate that the variability that the reviewer pointed out in the screen, particularly with respect to SKP2-C25 is likely to reflect both this limited range of efficacy and the fact that we do not assess actual compound concentration prior to a screen. For example, SKP2-C25 was provided at 10mM, but we have subsequently found it to be poorly soluble at this concentration. Therefore, we discuss the limitations of executing a screen at a single compound concentration.

Our dose response tests did, however, cast doubt on the validation of NU9056 as a hit (new Figure 6), whereby it did not show efficacy in dose response tests at any concentration, but did show a region specific effect in an independent test following the screen (Figure 7). Therefore, we classify this as a non-validated hit, which would not be pursued further, and in effect does represent a false positive.

We now also include a post hoc analysis of our screen that tests the premise that a predicted n=9 would have had the capacity to detect subtle changes in oligodendrocyte number. We find that lowering this n to either 6 or 3 would have led to identification of both false positive hits, and, more concerningly precluded identification of validated hits, thus returning false negatives (Supplementary file 4). We also discuss the issues of reproducibility and screen throughput in our new Discussion section.

We also provide further characterization of the effects of SKP2-C25, including time-lapse imaging based analyses, which demonstrate that the compound increases OPC proliferation in the zebrafish spinal cord (new Figure 4 and Figure 4—figure supplement 1 and Video 6).

As suggested by the reviewer, we have tested our top hit in a mammalian context, using a rat in vitro culture system, and show that splitomicin treatment increases myelinating oligodendrocyte number (new Figure 9).

Therefore, our screen identified compounds with validated effects on the oligodendrocyte lineage, including across species.

3) Why do various compounds targeting common molecules in similar manners (i.e. inhibiting sirtuin 1 and 2) have profoundly different effects on oligodendrocyte formation in this screen?

We discuss the fact that further studies are required to deconvolve the target(s) through which the various compounds identified in our screen function. For example, it is possible that splitomicin mediates its effects on either sirtuin 1 or sirtuin 2, or potentially through a balance of both. Different compounds that hit similar targets, may do so with different affinities for one or other of the shared targets. Indeed, it is also possible that the effective compounds mediate their functions on additional unknown targets.

4) A distinct disadvantage to this screening protocol as compared to others that have been previously published (e.g. Deshmukh et al., 2013; Mei et al., 2014) is that there is no method for discerning how each compound is affecting oligodendrocyte formation (e.g. directly acting on OPCs to cause an increase in OPC differentiation). Different diseases disrupt different aspects of oligodendrocyte formation and elucidating on which aspect of oligodendrocyte formation a compound is acting is important for discovering compounds useful for distinct diseases. As it stands, this manuscript does not provide substantial insights into oligodendrocyte biology. How do the identified compounds alter oligodendrocyte lineage cell biology? Visualizing both OPCs and oligodendrocytes could be a method by which the authors could provide some additional information on the mechanism of action of a compound. Which cell type(s) do they act on?

This is an important point, particularly with respect to triaging compounds for effects on different aspects of oligodendrocyte lineage development. We show here that both SKP2-C25 (new Figure 4—figure supplement 1 and Video 6) and splitomicin (new Figure 9) promote oligodendrocyte proliferation. We also discuss how combining further complementary studies in both zebrafish and cellular platforms will continue to provide important information on the cellular mechanisms of compound function.

Reviewer #3:[…] By fishy standards, the throughput of the screening example of only a few hundred compounds is low- only 175 compounds at a single dose vs. typical zebrafish screens these days of 1000s or even 10K compounds, but this is reasonable proof of principle for this kind of high-resolution imaging screen. The example of myelination is a well-chosen one to show off the capabilities, as such a screen would otherwise be a challenge. What might have been nice is a John Henry vs. the locomotive kind of comparison- i.e. how quickly could automated counters do the same screen, and what kinds of screening throughputs might now be possible and over what timescales? This might have really made the power of this approach clear.

We have now included a comparison of the speed of fully automated image analysis, versus a semi-automated approach wherein one manually corrects any errors in identification of cells versus a fully manual quantitation of cell number in 3D data stacks. This is now included in subsection “Automated quantification of oligodendrocyte cell number”. In addition, we have included more extensive discussion on how to best scale up screen throughput using this system in our new Discussion section.

Another weakness is that the manuscript is quite light on the biology- the compound hits haven't provided a clear biological insight as yet, nor even a clear target for further study. But I understand that eLife publishes technical/methods/resource papers and this would be a good example of that.

We have now added data showing that SKP2-C25 increases the proliferation of oligodendrocyte progenitor cells (OPCs) in zebrafish (new Figure 4—figure supplement 1 and Video 6) and that splitomicin promotes oligodendrocyte progression by increasing the proliferation of mammalian OPCs in culture (new Figure 9).

We thank the reviewer for their suggestion to re-submit our manuscript in the Tools and Resources category and agree that this is more appropriate.

[Editors' note: further revisions were requested prior to acceptance, as described below.]

Reviewer #1:[…] 1) The authors have used a dose response assay on their top four hits and validated three of the compounds. One of these hits NU9056 was no longer able to increase cell number, although it did still show a localized effect in a different assay. The authors attribute the variability seen between different experiments in part to the short window of action for the compounds thus highlights the importance of having good controls and finding optimum doses and this is now discussed in the text. I think one of the issues here is that the control compound is still being validated and isn't optimal in all experiments and therefore is effectively an extra hit compound. The authors do discuss this issue in their response. I also wonder if the variability seen is linked to the order of the compounds when being imaged and the time taken. The authors have discussed varying the order of the compounds imaged on the VAST in response to a point made previously and it would be of interest to know if there is any correlation.

The reviewer is correct in suggesting that the length of each treatment run is important, as one will certainly see larger variability in longer screening run, given continuous development of the animal. Indeed, the length of screening run in which our positive control compound represented a false negative was two times longer (8 hours) than when it was identified correctly as a hit (4 hours). This is a very astute point by the reviewer and we now refer to this in both the Results section and Discussion section.

2) To further characterize the action of the hit compounds the authors have used live imaging to show the control compound has increased proliferation. As this is quite a generic effect, it would be worth to note if this is specific to the OPCs or if there was an increase in all olig2:GFP and sox10:mRFP cells.

The density of olig2:GFP cells makes their precise quantitation from the data to hand very difficult. We do not see any general effects on animal size suggestive of general effects on proliferation but cannot exclude the possibility that compounds may exhibit effects on other cell types. We now note this in our Discussion section and underline the need to carry out cell-type specific genetic analysis of targets function (subsection “Novel regulators of oligodendrocyte formation”).

One of the hits, Splitomicin was also shown to increase MBP positive cells in rat oligodendrocyte cultures, further validating this hit compound in a different species and a different experimental assay as requested by one of the reviewers and this has improved the manuscript.As there are just three strong hits it would be of interest to know if these two different approaches (live imaging, rat oligodendrocytes) had been tested with the other compounds (and control). While I think that having the data for one compound validates the approach and is fine for this manuscript, it might be worth mentioning if these experiments with the other compounds are planned or if they were inconclusive.

We have only carried out the analyses on the respective compounds as shown to date, and we further underscore in the text the need to explore deeper analysis of all compounds in the future.

Reviewer #2:The authors have addressed most of the previous comments, however, there are a few remaining points to be addressed:

*1) Numerous times throughout the text, the authors state that SKP2-C25 and splitomicin promote OL lineage progression into OLs through increasing OPC proliferation (new data). Could the authors please elaborate on their suggestion that promoting OPC proliferation (where there are minimal asymmetric cell divisions that generate OLs (Hughes et al., 2013) could cause an increase in OL numbers? Known pro-proliferative compounds such as PDGF-AA are well-documented to decrease OL formation* in vitro*. In any case, proliferation does not necessarily promote lineage progression.*

We agree with the reviewer that promoting oligodendrocyte proliferation does not necessarily promote lineage progression. We clarify further in the text, why increasing OPC proliferation can lead to an increase in myelinating oligodendrocyte number. In subsection” Validation of automated analysis to identify changes in myelinating oligodendrocyte 216 number” we explain “We have previously shown that oligodendrocytes differentiate by default in zebrafish to mediate the early myelination observed in the larval spinal cord (Almeida et al., 2018). Therefore, upon an increase of OPC proliferation following SKP2-C25 treatment, there is a larger pool of OPCs that differentiate by default, leading to the observed increase in myelinating oligodendrocytes.”

In subsection “Automated analysis of myelination”, regarding the mammalian in vitro data, we note that “We saw an increase in the number of Ki67-expressing, olig2-expressing cells, upon splitomicin treatment (Figure 9), indicating that the observed increase in myelinating oligodendrocyte number was due to increased proliferation and subsequent differentiation of a larger pool of OPCs.”

We have revised the text to ensure that we do not suggest anywhere that an increase in OPC proliferation promotes oligodendrocyte lineage progression and thank the reviewer for this point.

2) In response to reviewer comments, the authors have increased the clarity of their data by including a discussion of the inconsistent reproducibility and false positives/negatives within the screen. This is important data-especially when considering the novel screening platform. In the spirit of transparency, the authors should report false negatives in the Results section. It is appropriate to report the non-significant trials of C646 and NU9056 when reporting their significant trials in detail (subsection “Chemical screen for compounds that regulate myelinating oligodendrocyte number”, C646; NU9056). The non-significant trial of SKP2-C25 could be reported here or elsewhere in the Results section.

We now include reference to the fact that SKP2-C25 failed to reach significance in one of its two screening runs in the Results section and note that the duration of the screening run was longer the time that it was not identified as a hit, likely due to the increased variability associated with longer screening runs on live animals.

3) Data from Figure 6 does not seem to be in Supplementary file 3. Additionally, please comment in the text or legend on how this data was obtained. Was it using the fully automated screening platform or another method?

We thank the reviewer for spotting this omission and we now include the new data accordingly and clarify in the Results section that this was indeed acquired on the automated screening platform.